# Spatial gene regulatory networks driving cell state transitions during human liver disease

Nigel L Hammond [1], Syed Murtuza Baker[1], Sokratia Georgaka [1], Ali Al-Anbaki[1], Elliot Jokl [1], Kara Simpson [1], Rosa Sanchez-Alvarez [1], Varinder S Athwal[1,2], Huw Purssell[1,2], Ajith K Siriwardena[1,2], Harry V M Spiers [3], Mike J Dixon[1], Leoma D Bere[1], Adam P Jones[1], Michael J Haley[1], Kevin N Couper[1], Nicoletta Bobola [1], Andrew D Sharrocks [1], Neil A Hanley [1,4,5], Magnus Rattray [1] & Karen Piper Hanley [1]✉

## Abstract

**Liver fibrosis is a major cause of death worldwide. As a progressive step in chronic liver disease, fibrosis is almost always diagnosed too late with limited treatment options. Here, we uncover the spatial transcriptional landscape driving human liver fibrosis using single nuclei RNA and Assay for Transposase-Accessible Chromatin (ATAC) sequencing to deconvolute multi-cell spatial transcriptomic profiling in human liver cirrhosis. Through multi-modal data integration, we define molecular signatures driving cell state transitions in liver disease and define an impaired cellular response and directional trajectory between hepatocytes and cholangiocytes associated with disease remodelling. We identify pro-fibrogenic signatures in non-parenchymal cell subpopulations co-localised within the fibrotic niche and localise transitional cell states at the scar interface. This combined approach provides a spatial atlas of gene regulation and defines molecular signatures associated with liver disease for targeted therapeutics or as early diagnostic markers of progressive liver disease.**

**Keywords** Liver Fibrosis; Spatial Transcriptomics; scRNAseq; scATACseq; Cell State
**Subject Categories** Chromatin, Transcription & Genomics; Digestive System; Methods & Resources

## Introduction

Chronic liver disease is increasing and a major cause of death worldwide (Williams et al, 2018). Although the liver has a remarkable capacity to regenerate after acute and transient injuries, repeated insult leads to progressive scarring (or fibrosis) and ultimately end-stage cirrhosis requiring transplant (Forbes and Newsome, 2016; Michalopoulos and Bhushan, 2021). Despite major advances in recent years, diagnosis tends to occur in advanced disease and clinically approved antifibrotic treatments are severely limited. A deeper understanding of the spatial complexity driving molecular and cellular mechanisms in human liver disease will be critical to identify early changes in pathophysiology leading to progressive disease and to develop novel therapeutics.

Pathological fibrosis underlies all forms of chronic liver disease and is associated with major changes to both the quantity and composition of the extracellular matrix (ECM) (Friedman and Pinzani, 2022). After acute liver injury, parenchymal cells (predominantly hepatocytes) regenerate and stimulate an inflammatory response (Michalopoulos and Bhushan, 2021). Subsequently, hepatic stellate cells (HSCs), the main ECM-producing cells of the injured liver, become activated and transdifferentiate into pro-fibrotic myofibroblasts (Athwal et al, 2017; Athwal et al, 2018; Hanley et al, 2008; Jokl et al, 2023; Martin et al, 2016; Pritchett et al, 2012). Iterative injury prolongs an impaired wound-healing response, resulting in the accumulation of excess ECM and the formation of fibrous scar tissue. Over time profound changes to the liver parenchyma and vasculature progress to cirrhosis, characterised by the formation of fibrous septa and regenerative nodules (Friedman and Pinzani, 2022). This unique microenvironment, involving complex interactions between multiple cell-types within the fibrotic niche, creates an impaired regenerative response and progressive disease (Forbes and Newsome, 2016; Friedman and Pinzani, 2022; Michalopoulos and Bhushan, 2021).

Advances in single cell and spatial technologies have significantly improved insight into the heterogeneity of both healthy and diseased liver; uncovering gene signatures associated with progenitor cells, zonation, immunological landscape and broad cellular response to disease pathogenesis (Aizarani et al, 2019; Dobie et al, 2019; MacParland et al, 2018; Matchett et al, 2024; Planas-Paz et al, 2019; Ramachandran et al, 2019; van den Brink et al, 2017; Marx, 2021; Chung et al, 2022; Guilliams et al, 2022; Hildebrandt et al, 2021; Hu et al, 2022a). While all are critical developments to the field, the paucity of chromatin accessibility data is limiting insight

[1]Faculty of Biology, Medicine & Health, Manchester Academic Health Science Centre, University of Manchester, Oxford Road, Manchester, UK. [2]Manchester University NHS Foundation Trust, Oxford Road, Manchester, UK. [3]Department of Surgery, University of Cambridge, Cambridge, UK. [4]College of Medical & Dental Sciences, University of Birmingham, Birmingham, UK. [5]University Hospitals Birmingham NHS Foundation Trust, Birmingham B15 2GW, UK. ✉E-mail: karen.piperhanley@manchester.ac.uk

on gene regulation in liver disease and the impaired regenerative response and disease states of parenchymal cells has been overlooked.

Here, we integrate ST, snRNA-seq and chromatin accessibility to characterise the spatial transcriptional landscape in human cirrhotic liver with comparison to healthy liver. In an initial discovery cohort, we combine multi-omic datasets from the same human cirrhotic tissue sample to uncover the cellular patho-architecture of diseased liver across three individual samples. This provided the initial analysis to define spatial molecular signatures of fibrotic scars, damaged parenchyma and impaired regenerative response at the scar interface for further validation in patient biopsies with severe liver fibrosis ($n = 5$) by ST or imaging mass cytometry (IMC). Through pseudo-temporal ordering, we show cell state transitions and directional trajectory between hepatocytes and cholangiocytes, potentially demar-cating impaired regeneration. We infer the gene regulatory networks (GRNs) differentiating these cell states and provide mechanistic understanding to gain novel insight into the gene signatures and regulatory programmes driving an impaired regenerative response during liver disease. Together, these data serve as a spatial atlas of gene regulation and altered cell state in human cirrhosis. They provide a framework to further interrogate molecular mechanisms under-pinning liver disease in larger disease orientated patient samples and provide a valuable resource to explore biomarkers and potential therapeutic targets in progressive liver disease (https://cellxgene.bmh.manchester.ac.uk/liver_disease/).

## Results

### ST resolves the spatial signature of fibrotic scars and altered cell state at their interface in human liver cirrhosis

Fresh, unfixed human tissue from three patients diagnosed with liver cirrhosis were collected to establish spatially resolved datasets using the 10X Visium platform (Appendix Information, Appendix Figs. S1A and S2, Table EV1). Tissue was cryosectioned and mounted onto capture areas of spatially barcoded Visium ST slides (Appendix Figs. S1A and S2), followed by hematoxylin and eosin (H&E) stain and imaging at high resolution before library preparation and sequencing (Appendix Fig. S1A). Following processing and mapping of the raw sequencing data, an average of 2986 genes were detected and mapped to 11,995 unique molecular identifiers (UMIs) per spot on the ST arrays. The entire dataset comprised a total of 8028 spots across four tissue sections (including one technical replicate; Appendix Fig. S3). Following manual annotation, histological landmarks were apparent demar-cating fibrous scar from the surrounding liver parenchyma. Although all patients were diagnosed with liver cirrhosis, as expected for human sample collection, the degree of fibrosis was noticeably varied, ranging from numerous fibrous septa (sample a) to extensive bridging septa encompassing regenerative nodules (sample b) and localised portal fibrosis (sample c) (Appendix Fig. S1B–D).

To explore gene expression underlying the cirrhotic patient samples, dimensionality reduction of spatial data was performed followed by unsupervised clustering and differential gene analysis (Appendix Information and Appendix Figs. S1 and S3). Initially, we

used $k$-means clustering to investigate spatial gene expression. Projecting spots and their clusters back onto the samples revealed spatial spots that could resolve fibrotic landmarks from the surrounding tissue (Appendix Fig. S1B–G). Localised regions of fibrosis were present in sample c and this was reflected by a single cluster (c2; orange) (Appendix Fig. S1D,G). However, in samples with more extensive scarring (samples a, b), several clusters were associated with fibrotic scars (Appendix Fig. S1B,C). Here, spatial clustering suggested heterogeneity within the scar, such that tissue at the scar interface was defined by clusters a2, b2 (orange) while the centre of the scar was mainly represented by clusters a4, b4 (red) (Appendix Fig. S1E,F). Interestingly, cluster b5 (purple) defined discrete locations towards the edge of the scar and was only present in the sample containing more extensive fibrosis (Appendix Fig. S1F). In contrast, $k$-means clusters broadly represented the functioning liver parenchyma by 1–2 large clusters per sample (a1, a3; b1; c1, c2) (Appendix Fig. S1B–D).

Although all patients had a clinical diagnosis of cirrhosis, to capture the most diverse cell-types and provide broad insight into spatial gene expression associated with fibrotic scarring we further interrogated sample b, displaying the most diverse pathology (Appendix Fig. S1), and compared analysis using both $k$-means and unsupervised Louvain clustering methods (Fig. 1A–D, Datasets EV1 and EV2). While $k$-means defined the spatial heterogeneity of fibrotic scars (Fig. 1A,B and Dataset EV1), Louvain clustering defined heterogeneity within the functioning liver parenchyma (b7, b8, b10) (Fig. 1C,D and Dataset EV2). In particular, cluster b9 (red) defined the tissue-interface with the fibrotic scar (b6; orange) potentially associated with damaged or impaired/regenerating parenchyma (Fig. 1C,D). For both analyses, we displayed differentially expressed genes (DEGs) per cluster as a heatmap, and annotated co-expression gene modules to spatial regions within the tissue (e.g. scar, interface, tissue; Fig. 1B,D).

Gene ontology (GO; Dataset EV3) analysis of $k$-means clusters identified *fibrosis*, *regulation of fibroblast proliferation* and *ECM-receptor interaction* terms within the scar interface cluster (b2; Fig. 1E) that spatially mapped to the periphery of the scar (Fig. 1I). As expected, pro-fibrotic markers and ECM components associated with activated HSCs were also evident and enriched in cluster b2 (*COL1A1*, *COL1A2*, *COL3A1*, *COL6A2*, *TIMP1*, *BGN*, *LAMC3*, *ADAMTSL2*, *SPHK1*, *AEBP1*, *CCN2*), with several targets co-expressed at the scar interface (Appendix Fig. S4A). Scar-associated cluster b4 was significantly enriched for *B cell signalling* and *B cell activation*, mapping spatially to the centre of the scar (Fig. 1F,J). Accordingly, marker genes of immune and activated B cells (*IGLC2*, *IGHG1*, *IGHG3*, *JCHAIN*, *MZB1*, *DERL3*, *S100A9*, *LSP1*) were enriched in this region, along with several mechano-sensitive components (*LMNA*, *TALGN*) (Fig. 1F; Appendix Fig. S4B). Although cluster b5 was the smallest cluster (28/1609 spots), these spots occupied discrete locations at the scar interface. GO analysis revealed enrichment for genes involved in *cholangiocarcinoma*, *tight junction assembly* and *oncostatin m* signalling, which also spatially mapped to discrete regions within and towards the edge of the scar, potentially indicative of ductal hyperplasia associated with severe liver disease (Fig. 1G,K). Markers associated with cholan-giocytes (*KRT19*, *KRT7*, *KRT23*, *EPCAM*, *AQP1*, *SPP1*, *CD24*) and chemokine ligands (*CXCL1*, *CXCL6*, *CXCL8*), were enriched both in cluster b2 and b5, localised at the scar interface (Fig. 1E,G,K; Appendix Fig. S4C).

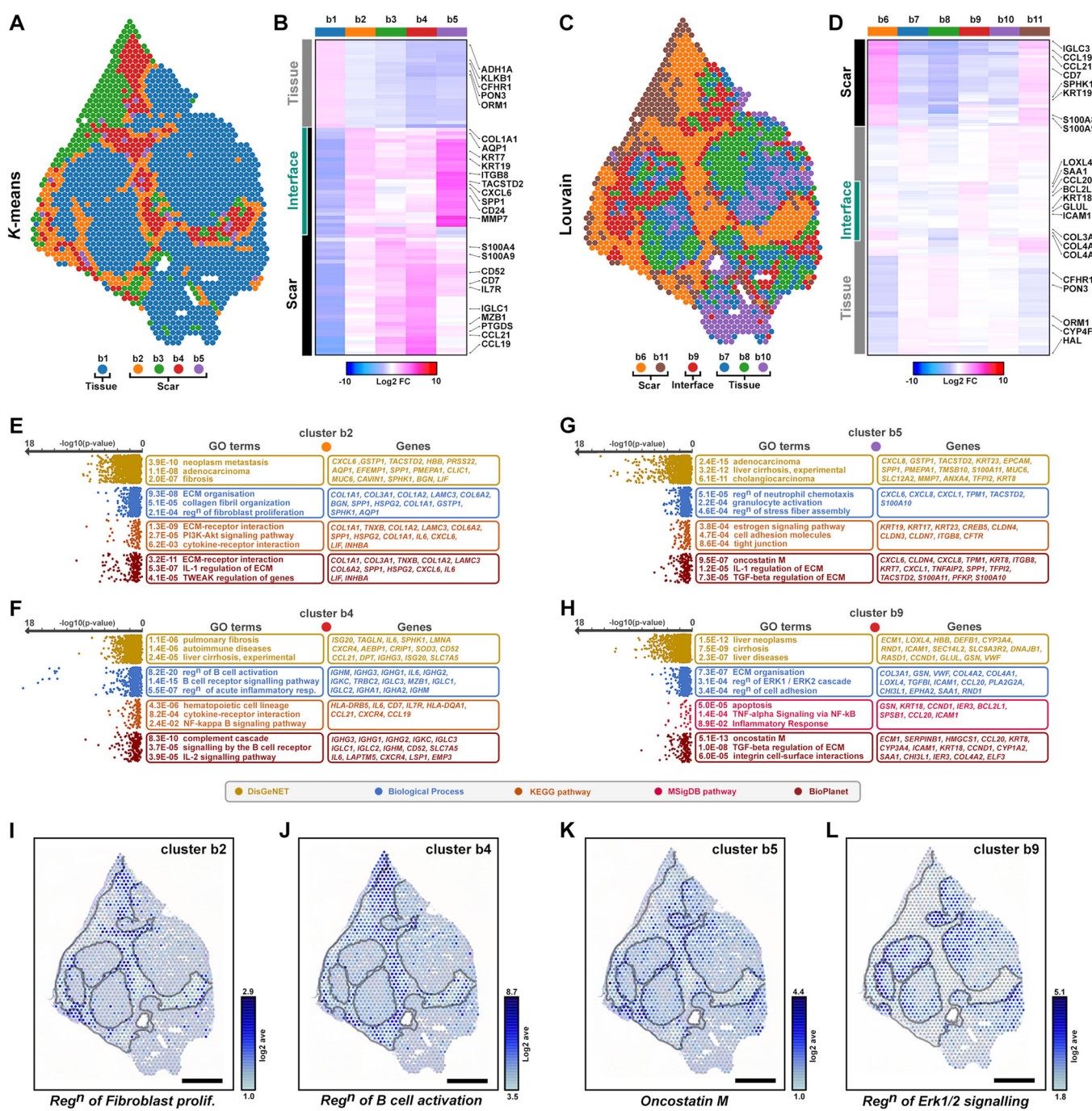

**Figure 1. Spatial signatures underlying fibrotic scar and interface with liver parenchyma.**

(A, B) Visualisation of spatial transcriptomic clusters for *k*-means (A) shows heterogeneity within fibrotic scar, identifies scar and interface clusters, with significant differentially expressed genes shown as a heatmap (B). (C, D) Louvain clustering shows heterogeneity within liver parenchyma (C), including a tissue-interface cluster and significant genes underlying them (D). (E–H) Gene ontology (GO) of significant genes underlying clusters, spots represent each GO terms and its −log10(*p*-value). (I–L) Spatial expression (log2 average UMIs) of gene modules underlying GO terms. Scale bars, 1 mm. Data information: In (B, D), heatmaps show differentially expressed genes (Z-score, log2 fold change).

Similarly, GO analysis of DEGs from Louvain clustering indicated enrichment of genes within cluster b9 (red; Fig. 1C,D) were involved in *ERK1/2 signalling, apoptosis, inflammatory response* and *integrin cell surface interactions* (Fig. 1H) spatially located at the scar interface (Fig. 1L). These regions also correlated with the spatial expression of pro-fibrotic ECM components (*LOXL4, COL3A1, COL4A1*), damage-induced inflammatory response proteins (*SAA1, SAA2, DEF1B*), apoptosis (*GDF-15, BCL2L1, TM4SF1*), and chemokine markers associated with hepatocyte regeneration (*CCL20, ICAM-1*) (Fig. 1H; Appendix Fig. S4D).

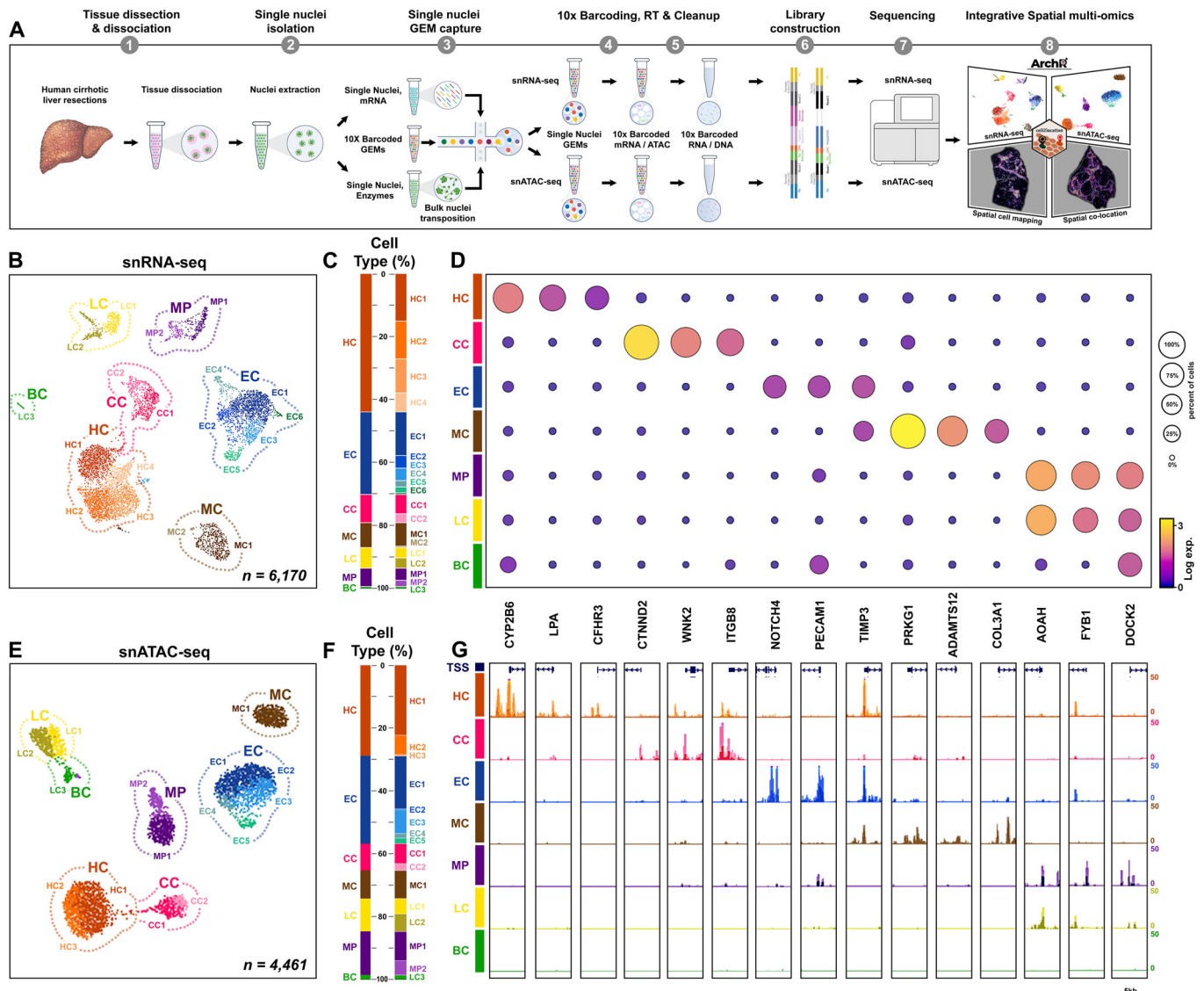

**Figure 2. Integrated single nuclei multi-omic analysis of human cirrhotic liver.**

(**A**) Schematic of 10X Chromium workflow to generate snRNA-seq and snATAC-seq libraries from cirrhotic liver resections, followed by sequencing and integrated spatial multi-omic analysis. (**B**) UMAP clusters show 7 discrete cell populations, which when re-clustered at high resolution reveal 19 sub-clusters within lineages (*n* = 6170 nuclei). (**C**) Stacked bar chart shows the proportion of nuclei (%) assigned to each cluster/sub-cluster. (**D**) Dot plot shows transcript expression and cell proportions of marker genes across cell types (log UMIs). (**E, F**) UMAP plot of snATAC-seq cell types (*n* = 4461 nuclei) with label transfer of sub-clusters from snRNA-seq data (**E**) and proportion of nuclei (%) within each cluster/subcluster (**F**). (**G**) Promoter accessibility of cell type marker genes (UCSC signal tracks) and correlation with gene expression (**D**). Data information: HC hepatocytes, CC cholangiocytes, EC endothelial cells, MC mesenchymal cells, MP macrophages, LC T lymphocytes, BC B cells.

Taken together, ST uncovered gene signatures underlying the pathophysiology in human cirrhotic liver tissue potentially indicating an impaired response to on-going disease. However, information on the cell-type contribution to these signatures was obfuscated by the current resolution of the spatial spots.

## Integrated spatial multi-omic analyses of human cirrhotic liver

To improve resolution and provide a refined spatial cell-type map, we jointly profiled the transcriptome (snRNA-seq) and accessible chromatin (snATAC-seq) from human cirrhotic liver at single

nuclei resolution (from sample b; Fig. 2A). From 6170 nuclei, an average of 2445 genes and 8368 UMIs per nuclei were detected for snRNA-seq (Appendix Fig. S5A,B). Following dimensionality reduction (McInnes et al, 2018) and low-resolution clustering, seven hepatic cell populations were identified (Fig. 2B,C). In combination with publicly available datasets (Aizarani et al, 2019; MacParland et al, 2018; Payen et al, 2021; Ramachandran et al, 2019), differential expression analyses identified markers for all major liver cell-types in our dataset, specifically hepatocytes (HC; 45%), endothelial cells (EC; 26%), cholangiocytes (CC; 9%), mesenchymal cells (MC; 7%), macrophages (MP; 6%), lymphocytes (LC; 6.5%) and B cell (BC; 0.5%) populations (Fig. 2B,C). From the

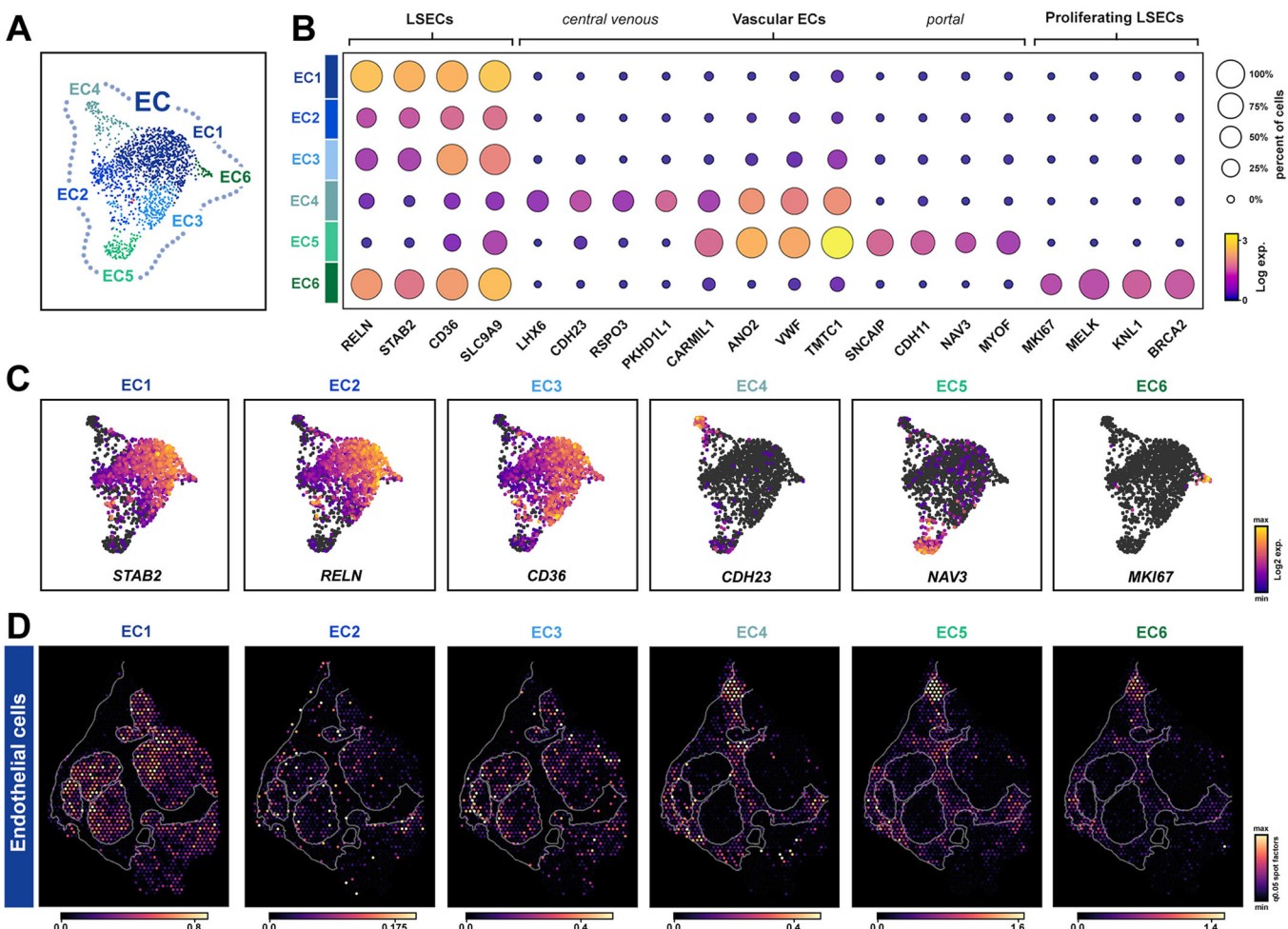

**Figure 3. Spatial heterogeneity of endothelial cell populations in cirrhotic liver.**

(A–C) UMAP sub-clusters (A), dot plots (B) and marker gene expression plots (C) showing heterogeneity of sub-populations of endothelial cells (EC). (D) Spatial deconvolution maps of EC sub-populations within cirrhotic sample b. Data information: In (A), each spot represents a single nuclei. In (C), marker gene expression (log2 UMIs) for sub-populations EC1-6. In (D), spatial spots show cell abundance (colour intensity) for each sub-cluster. LSECs, liver sinusoidal endothelial cells.

entire dataset we further uncovered cellular heterogeneity to define 19 subclusters within the seven discrete cell populations (Fig. 2B,C; and Dataset EV4).

Simultaneously, we profiled chromatin accessibility from the same human cirrhotic liver sample. From 4461 nuclei, we detected 199,455 regions of accessible chromatin which using unsupervised clustering resolved to seven discrete cell populations (Fig. 2E,F; Appendix Fig. S5C,D and Dataset EV5). To define cell-types based on chromatin accessibility profiles, we leveraged our annotated snRNA-seq data and performed label transfer using Seurat (Stuart et al, 2019). Integration of paired modalities confirmed that all major liver cell-types were present in both datasets (Fig. 2C,F). Furthermore, transfer of snRNA-seq subcluster annotations confirmed the majority of cell states (16 out of 19 subclusters) were represented in the snATAC-seq dataset (Fig. 2B,C,E,F). Combined analysis of cell-type markers of gene expression and promotor accessibility across both modalities were highly correlated, validating our integrative single nuclei multi-omic approach (Fig. 2D,G; Appendix Fig. S6A and Datasets EV4 and EV5).

## Spatial heterogeneity of non-parenchymal cell populations in human cirrhotic liver

Next, we explored snRNA-seq gene signatures underlying the heterogeneity of non-parenchymal cell types in cirrhotic liver, and separately integrated our data with similarly prepared healthy human liver (Andrews et al, 2022) for comparison with spatial validation in patient biopsy samples (Appendix Figs. S7 and S8). ECs were identified by the marker genes *PTPRB*, *LDB2*, *NRG3*, *NOSTRIN* and *AKAP12*, and were further characterised into six distinct sub-clusters (EC1-6) (Fig. 3A). EC1, 2 and 3 represented over 80% of the total ECs and were annotated as liver sinusoidal endothelial cells (LSECs; a specialised microvascular EC) using the marker genes (*RELN*, *STAB1*, *STAB2*, *MRC1*, *CD36*) (Fig. 3B,C). Although clusters EC4 and EC5 were well separated in UMAP, they uniquely co-expressed several genes (*ANO2*, *VWF*, *CARMIL1*, *BMX*, *CDH23*) indicating a vascular EC phenotype. Specifically, EC4 was enriched for WNT components and other markers (*RSPO3*, *WNT9B*, *WNT2*, *LHX6*, *CELF4*) suggesting this population

were central venous ECs, while EC5 marker genes (*SNCAIP, CDH11, MYOF, NAV3, TNFRSF11A*) identified these cells as portal ECs (Fig. 3A–C). Cluster EC6 consisted of just 31 cells and expressed markers of proliferating LSECs (*MKI67, PCNA, RELN*). In healthy liver, we observed overlap of our EC1-3 populations with LSECs and portal ECs in the UMAP, whereas EC4-6 were more distinct from healthy cells, potentially indicating disease-specific gene signatures underlie these subclusters (Appendix Fig. S9A,B). In addition, many of these markers were enriched in comparison to ECs from healthy liver (Appendix Figs. S9C,D and S10A,B,G,H). Spatial mapping of cirrhotic EC subpopulations revealed the distribution of LSECs (EC1-3) within the main tissue architecture, while EC4-6 were differentially enriched within regions of the scar, suggesting a shared profibrotic signature (Fig. 3D). Of interest, we have previously described NAV3 (from EC5) in activated pericytes (kidney specific myofibroblasts) during chronic kidney disease (Raza et al, 2021) highlighting potentially shared mechanisms of fibrosis. To confirm our profibrotic EC signature, a similar profile was evident following spatial mapping in our fibrotic validation cohort (*n* = 4) using Visium ST compared to the gene signature distribution in heathy liver (Appendix Figs. S8, S9E,F and S10A,B,G,H).

MCs displayed markers representative of resident liver fibroblasts, Hepatic Stellate Cells (HSCs), which in response to liver injury can transdifferentiate into profibrotic activated myofibroblasts (activated HSCs). MC1 comprised the majority of MCs (Fig. 4A) and was enriched for *PDGFRA* and *PDGFRB*, alongside HSC marker genes (*LAMA2, ADAMTS12, ADAMTSL1, RELN, VIPR1, NAV3*) (Fig. 4A,B,G). Consistent with previous studies(Andrews et al, 2022; Dobie et al, 2019; Ramachandran et al, 2019), comparison to healthy single cell data showed our MC1 population displayed both markers of quiescent qHSCs present in healthy liver (*RBP1, LRAT, SPARCL1, LHX2*) and activated HSCs (aHSCs) responsible for tissue damaging ECM deposition (*COL1A1, TIMP1, ACTA2, NCAM, SPARC*; Appendix Fig. S11A–D). MC2 comprised only 17 cells and were *PDGFRA-/PDGFRB+*, with parallels to a vascular smooth muscle cells (vSMCs) origin. As expected, differential gene analysis revealed marker genes of vSMCs (*MYH11, MCAM, NOTCH3, PPARG, CNN1*) enriched in MC2 (Fig. 4A,B,G). In line with their distribution from healthy HSC populations in single cell comparative analysis, spatially, MC1 and MC2 mapped to discrete regions within the scar in fibrotic tissue (Fig. 4J; Appendix Fig. S11E,F) with a similar profile to EC4-6 (Fig. 3D). In healthy liver profibrotic gene signatures were more evenly dispersed and critically of a much lower expression level overall compared to their distribution within areas of scarring in the fibrotic validation cohort (Appendix Figs. S10A,C,G,H and S11E,F).

From previous studies characterising immune cell populations in health and fibrotic liver (Andrews et al, 2022; Guilliams et al, 2022; MacParland et al, 2018; Ramachandran et al, 2019), we recognise that alternative parallel approaches may be beneficial to fully capture all cell populations (e.g. use of single cell rather than single nuclei techniques and/or cell sorting). However, from our data, we identified a broad cluster of MPs which further resolved to two distinct subpopulations (Fig. 4C). MP1 was defined by *CD163+/MARCO+* cells and together with marker genes (*EXT1, LILRB5, CD169, NDST3, MSR1, LYVE1*) represented the liver resident macrophages, Kupffer cells (KCs) (Fig. 4D,H). In contrast,

MP2 was characterised by *MARCO-/MNDA+* cells and with marker genes (*AFF3, RTN1, FLT3, CCSER1, XYLT1, CIITA*) identified as injury-recruited inflammatory macrophages, including granulocytes and dendritic cells (IMs; Fig. 4D,H) (Aizarani et al, 2019; MacParland et al, 2018; Ramachandran et al, 2019). In keeping with these data, gene signatures of MP1 localised within the liver tissue compartment indicative of liver resident KCs, whereas MP2 was spatially distributed within the scar indicative of injury (Fig. 4K). Although there was some overlap in all macrophage cell populations compared to healthy liver, gene expression appeared enriched in cirrhotic samples particularly in the case of the inflammatory (MP2) signatures. As validation, MP2 signatures were spatially located within the scar of patient biopsy samples compared to the broad profile in healthy tissue (Appendix Figs. S10A,D,G,H and S12).

T and B lymphocyte populations resolved to three distinct clusters (LC1, LC2 and LC3; Fig. 4E). Marker gene analyses detected T cell signatures in LC1 (*IL7R, THEMIS, INPP4B, CAMK4, PBX4*) while LC2 population was associated with cytotoxic natural killer T (NKT) cell signatures (*KLRF1, NCAM1, NCR1, LINGO2, CEP78*) (Fig. 4F,I). B cells (BC) represented the smallest cluster in LC3 (*IGHA1, TENT5C, LRRK1*) (Fig. 4F,I). Spatial mapping of T and B lymphocytes revealed their enrichment within the scar of our experimental and validation fibrotic samples compared to healthy tissue, indicative of their role during chronic liver disease (Fig. 4L; Appendix Figs. S10A,E,G,H and S13). Together, these data characterised the gene signatures underlying disease-associated non-parenchymal cell populations.

## Cellular heterogeneity of parenchymal cell populations in human cirrhotic liver

To explore the gene signatures underlying parenchymal subclusters, we initially defined the CC population by broad cell-specific marker genes (*PKHD1, CTNND2, BICC1, ANXA4, DCDC2*). Further heterogeneity was revealed by resolving to two distinct sub-clusters (CC1, CC2; Fig. 5A). Differential expression analysis showed mature CC markers (*KRT7, KRT19, CFTR, AQP1, HNF1B*) were more highly expressed in CC2, whereas interestingly CC1 demonstrated a gene signature with similarities to HCs (*GPC6, APOB, PPARGC1A, MYLK, MLXIPL, FGF13, FGFR3, SULF1, RUNX1*) (Fig. 5B). Exploring these markers, *SULF1* is normally restricted to the developing fetal liver but can be upregulated in injured liver, potentially from attempted regeneration (Graham et al, 2016). Similarly, FGF signalling is important for hepatocyte regeneration (Padrissa-Altes et al, 2015) and we noted several FGF components enriched in CC1 and HCs (*FGF13, FGFR3*). Although the function of FGF13 in liver is less well understood, studies have indicated its potential as part of a seven-gene signature for HCC progression (Liu et al, 2020). As a result, the profile of CC1 suggested an immature CC population that are HC-like, potentially indicative of an attempted regenerative response.

HCs represented the largest cell-type cluster with 45% of sequenced nuclei. Following re-clustering, this population resolved to 4 sub-clusters (HC1-4) with HC1 isolated from HC2/3 (Fig. 5A). We noted HC4 had much lower UMI counts (Appendix Fig. S5) and was not present in snATAC-seq data, therefore these cells were not characterised further. Hepatocyte zonation in the human liver has been well documented (Droin et al, 2021; Halpern et al, 2017),

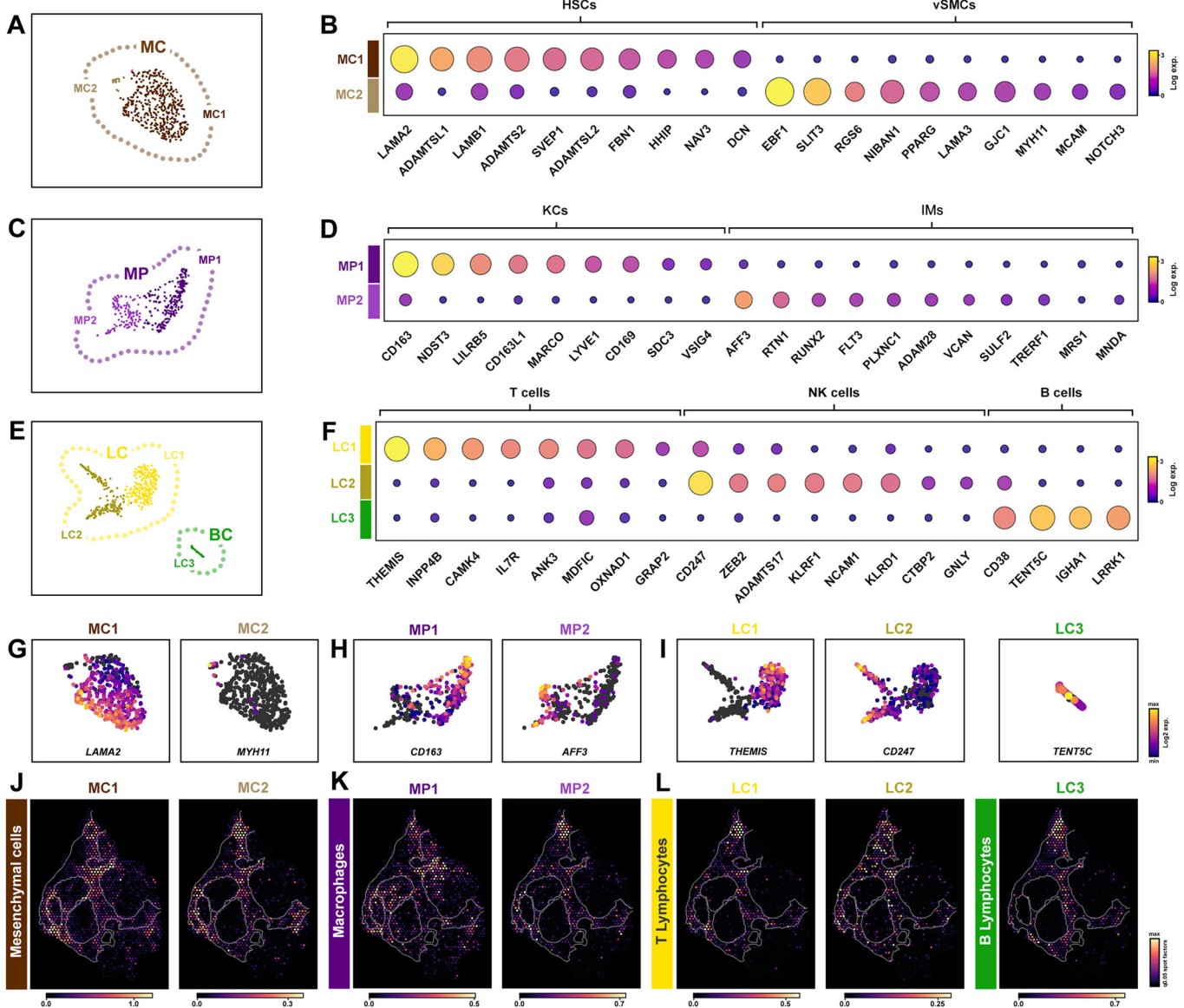

**Figure 4. Spatial heterogeneity of non-parenchymal cell populations in cirrhotic liver.**

(A–I) UMAP sub-clusters (A, C, E), dot plots (B, D, F) and marker gene expression plots (G, H, I) showing heterogeneity of mesenchymal (MC; A, B, G), macrophage (MP; C, D, H) and immune (LC, BC; E, F, I) cell sub-populations. (J–L) Spatial deconvolution maps of MC (J), MP (K) and immune (L) cell sub-populations within cirrhotic sample b. Data information: (A, C, E, G–I) Each spot represents a single nuclei. (G–I) Marker gene expression (log2 UMIs) for non-parenchymal cell sub-populations (J–L) Spatial spots show cell abundance (colour intensity) for each sub-cluster (cell2location). HSC hepatic stellate cells, vSMCs vascular smooth muscle cells, KCs kupffer cells, IMs inflammatory macrophages, LC lymphocytes, BC b cells.

including gradients of gene expression from periportal (PP) to pericentral (PC) regions which reflect distinct metabolic functions across the liver lobule. Evaluation of known PP (*HAL, MRC1, SDS*) and PC (*CYP3A4, CYP2E1, GLUL*) markers identified opposing gradients of gene expression with HC2/HC3 enriched for PP/PC markers, respectively (Fig. 5B,C). In keeping with their characteristic zonation profiles, spatial maps showed gene enrichment signatures for periportal (PP; HC2) hepatocytes were associated with portal bridging fibrosis (Fig. 5D). Whereas, pericentral (PC; HC3) hepatocytes occupied complementary regions within the main tissue (Fig. 5D).

HC1 displayed PP/PC zonation markers, similar to HC2/HC3, ruling out zonation as the underlying cause for segregation of this cluster (Fig. 5B,C). However, further characterisation of HC1 highlighted several hepatocyte signature genes that were not present (*LINC01554, MAGI2, TRIM55, ZEB1, ZIC1, HEPACAM, TRPS1*) compared to HC2/HC3 (Fig. 5B,C). Significantly, downregulation of many of these genes influences cell proliferation, migration and fate (Drapela et al, 2020; Hao et al, 2020; Hu et al, 2007; Xun et al, 2010; Zhe et al, 2015) while reduced expression of *LINC01554* is associated with advancing fibrosis in NAFLD patients (Ryaboshapkina and Hammar, 2017). Further analyses revealed HC1 uniquely shared

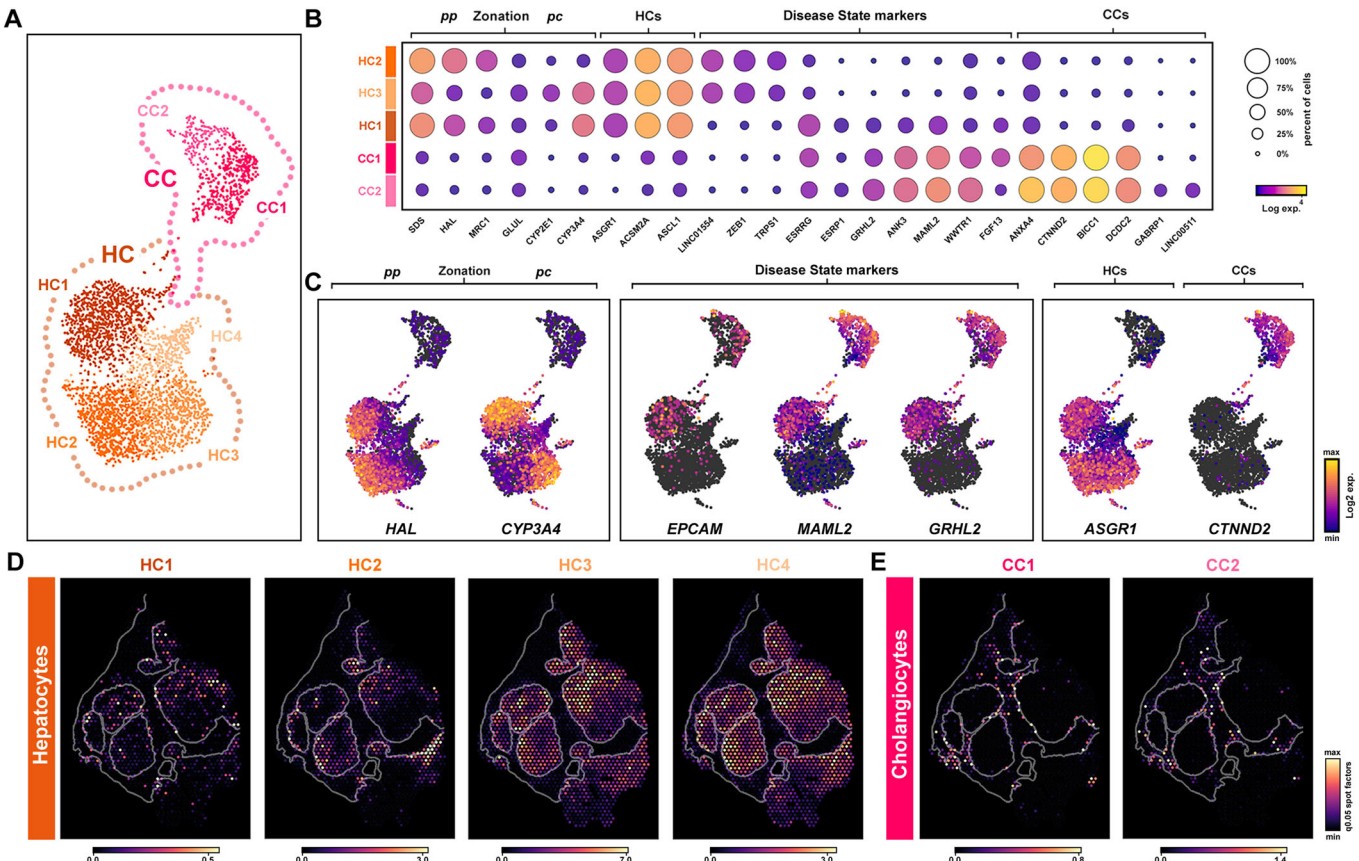

**Figure 5. Spatial characterisation of parenchymal sub-populations in cirrhotic liver.**

(A) UMAP plot of parenchymal cells show hepatocyte (HC; HC1-HC4) and cholangiocyte (CC; CC1, CC2) sub-populations. (B, C) Dot and UMAP expression plots (log2 UMIs) show marker genes of HC zonation, HC/CC broad cell types and disease-associated cell states. (A, C) Each spot represents a single nuclei. (D, E) Spatial deconvolution maps of HC (D) and CC (E) sub-populations within cirrhotic sample b. Data information: In (A, C), each spot represents a single nuclei. (D, E) Spatial spots show cell abundance (colour intensity) for each sub-cluster (cell2location). PC pericentral, PP periportal.

transcriptional signatures with CCs (*ARHGEF38*, *GRHL2*, *MAML2*, *ESRRG*, *ESRP1*, *ZNF83*, *EPCAM*), suggesting a transition between parenchymal cell states (Fig. 5B,C). Accordingly, GRHL2 is implicated in epithelial cell fate and CC differentiation (Tanimizu et al, 2014; Tanimizu et al, 2013), while ESRP1/2 are epithelial splicing factors involved in lineage differentiation programmes and HC regeneration (Bangru et al, 2018; Bhate et al, 2015; Hyun et al, 2020; Warzecha et al, 2009). Although EpCAM is not usually expressed in mature HCs, it is associated with hepatobiliary progenitors and nascent HCs in diseased liver (Aizarani et al, 2019; Athwal et al, 2017; Segal et al, 2019; Yoon et al, 2011). Significantly, *EPCAM* was uniquely expressed in HC1 identifying this cluster as potentially disease-associated, reminiscent of our previous work (Athwal et al, 2017) (Fig. 5B,C).

In comparison to healthy liver, HC1 and part of HC2 were separate from healthy hepatocytes populations with discrete gene expression. Similarly, although there was some overlap with the CC1 and CC2 cell populations to healthy cholangiocytes, gene expression aligned to HC1 was evident (Appendix Fig. S14A–F). Moreover, spatially, HC1 and CC1 subpopulations were enriched at the scar-interface, while gene signatures underlying CC2 were

localised further within the scar in our experimental tissue and validation patient fibrotic biopsies (Fig. 5D,E; Appendix Figs. S10A,F,G,H and S14G,H). Taken together, snRNA-seq identified atypical HCs (HC1) that shared transcriptional signatures with immature CCs (CC1), indicating a population of disease-associated parenchymal cells potentially involved in an impaired regenerative response during progressive fibrosis.

Although direct comparisons of gene signatures between different spatial technologies is more divergent (i.e. frozen used for the initial cohort 'v' FFPE in the validation cohort) there are likely similar themes. To provide insight into common fibrotic gene expression programmes, we compared the top 100 differentially expressed genes in 'scar associated' Louvain clusters from all spatial samples (initial and validation cohorts) and compared expression across our validation cohort (Dataset EV6). The most prevalent fibrotic gene signatures evident and enriched were from the MC populations ranging from moderate fibrosis (samples D and E) through to severe fibrosis (samples F and G). In addition, there was a noticeable alteration and recruitment of inflammatory cell gene signatures in severe fibrotic biopsies contributing to the overall niche driving fibrosis.

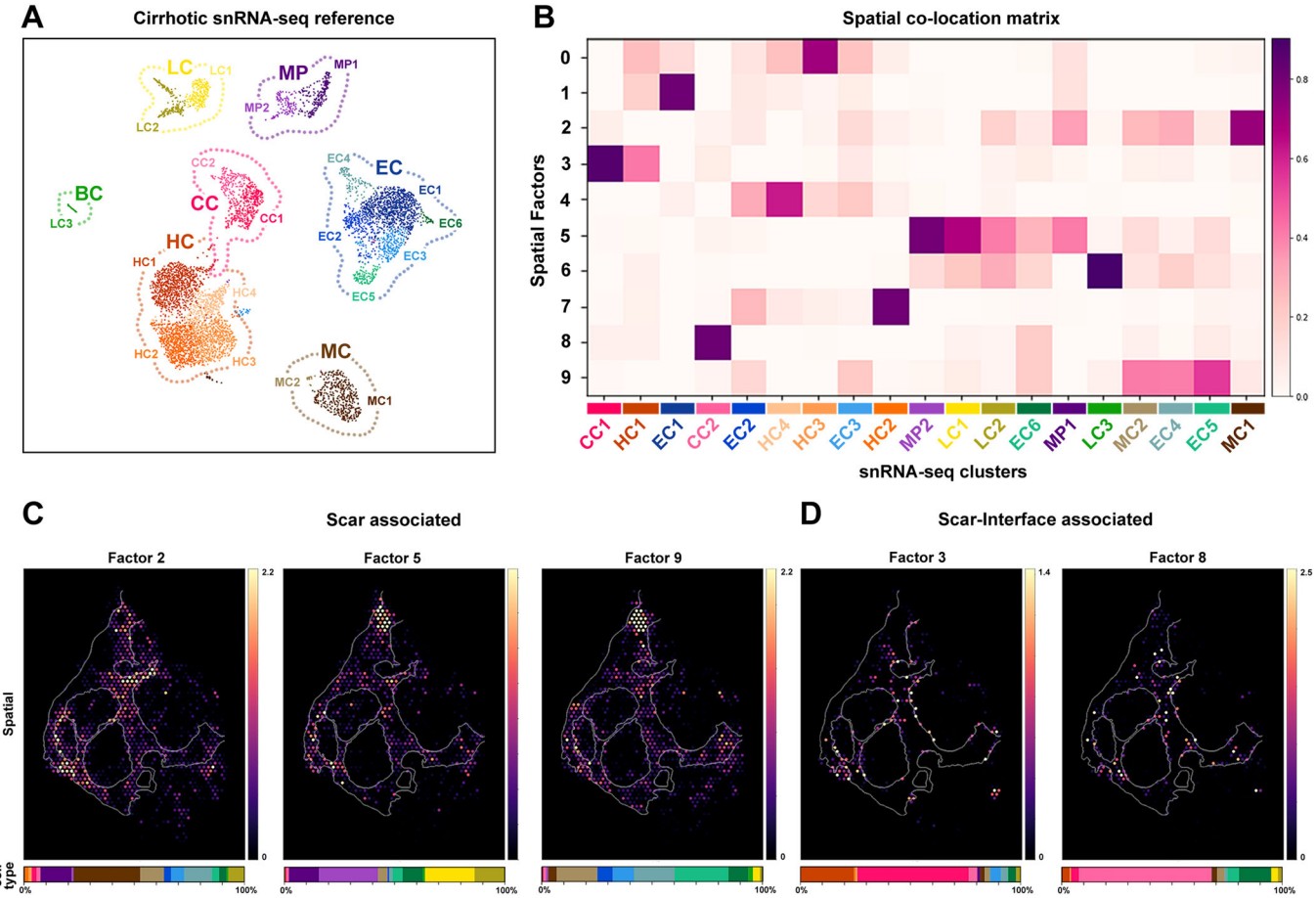

**Figure 6. Spatial deconvolution of the fibrotic niche.**

(A) Reference gene signatures from snRNA-seq sub-populations were used to deconvolute multi-cell ST spots (cell2location). Each spot represents a single nuclei. (B) Co-location analysis for sample b using non-negative matrix factorisation (NMF) shows 10 factor maps (0–9) and the proportion of cell sub-populations to each spatial factor. (C, D) Spatial deconvolution maps of scar-associated (2, 5 and 9; C) and scar-interface associated (3 and 8; D) factors reveal spatial co-location of cell sub-populations in cirrhotic sample b. Data information: In (C, D), spot intensity depicts estimated cell abundance and corresponding bar charts show percentage of cell subpopulations which represents each factor. Hepatocytes (HC), endothelial cells (EC), cholangiocytes (CC), macrophages (MP), mesenchymal cells (MC), T lymphocytes (TC), B lymphocytes (BC).

## Spatial deconvolution of the fibrotic niche

To increase our understanding of the spatial heterogeneity in cirrhotic tissue, we combined the strengths of ST and snRNA-seq gene signatures (Figs. 2A and 6A) using cell2location (Kleshchev-nikov et al, 2022) to deconvolute multi-cell ST spots (Dataset EV7). Using non-negative matrix factorisation (NMF), we clustered cell profiles (Fig. 6B) and inferred the co-location of cell subpopulations. Spatial deconvolution of 19 cell subpopulations (Appendix Figs. S15–S18) mapped to 10 co-location factors (sample b; Factor 0–9; Fig. 6B), revealing several cell subpopulations spatially enriched within cirrhotic tissue (Fig. 6B–D; Appendix Figs. S16 and S18). Significantly, we focussed on spatial maps that were scar associated (e.g. Factors 2, 5 & 9 in sample b) and located at the scar-interface (e.g. Factors 3 & 8 in sample b) (Fig. 6C,D), including a comparison of the scar associated and interface associated factors across all experimental samples (samples a–c; Appendix Fig. S19).

Factor 2 was comprised of HSCs (MC1), a smaller contribution from resident macrophages (MP1) and a subpopulation of ECs (EC4)

(Fig. 6B,C). To uncover the spatial signature of these scar-associated subpopulations, we analysed DEGs underlying Factor 2 spots and linked expression to cell-type profiles from the snRNA-seq reference (Appendix Fig. S20A,B). As expected, collagen transcripts (COL1A1, COL1A2, COL3A1, COL6A2) were significantly enriched in the scar alongside secreted ECM components and other profibrogenic markers indicative of aHSCs (MC1; SPARC, AEBP1, TAGLN, FSTL3, LAMC3, RBP1, ADAMTSL2, IGFBP7; Appendix Fig. S20A–C). Spatial deconvo-lution identified novel HSC genes, QSOX1 and RAMP1 (Appendix Fig. S20B,C) (Andrews et al, 2022; Payen et al, 2021) as part of the profibrotic niche.

Several immune cell populations were spatially co-located within the fibrotic scar, represented by Factor 5 (Fig. 6B,C), and comprised resident macrophages (MP1), infiltrating inflammatory macrophages (IMs; MP2) and lymphocyte populations (LC1, LC2). Although these immune cells resolved to a unique factor, they co-located to similar regions of the scar occupied by aHSCs (Fig. 6B,C). Spatial DEGs of Factor 5 highlighted markers of inflammatory macrophages (HLA-DQA1, HLA-DRB5, LYZ, VCAN, CD52, CD74) and T cells (IL7R, TRBC2, CD7) enriched within

the fibrotic niche (Appendix Fig. S20D–F). Following injury, the pleiotropic cytokine macrophage migration inhibitory factor (MIF) is secreted and signals via cell surface receptor/co-receptors CD74/CD44 expressed on macrophages and lymphocytes (Wirtz et al, 2021). Together, MIF-CD74/CD44 signalling components were spatially enriched at the scar/tissue interface, highlighting tissue regions undergoing inflammation-driven repair orchestrated by multiple immune cell-types (Appendix Fig. S20F).

Interestingly, Factor 9 demonstrated the spatial expression of vascular-associated EC subpopulations (EC4, EC5), also occupying similar hotspots within the fibrotic scar (Fig. 6B,C). Spatial DEGs revealed the scar-associated expression of pericentral ECs (EC4; *PTGDS, TAGLN, ADAMTS1, ADAMTS4, TNXB*) and periportal ECs (EC5; *PLVAP, VIM, EMP1, VWF, LTBP4*) (Appendix Fig. S20G–I). In agreement with our spatial data, PLVAP+ ECs have been similarly verified as scar-associated and expanded in cirrhosis (Ramachandran et al, 2019).

Spatial DEGs underlying Factors 3 and 8 showed an enrichment of cholangiocyte markers (*KRT19, KRT7, CFTR, AQP1*), but also genes characteristic of liver progenitor-type cells (LPCs; *TACSTD2, CD24, EPCAM, SOX9, PROM1, SPINT1, CLDN4/7, KRT23*) (Appendix Fig. S21A,B) (Segal et al, 2019). Similar to our previous work localisation of SOX9 and EPCAM in human liver development and disease-associated hepatocytes (Athwal et al, 2017), SOX9 and TACSTD2 (also known as TROP2) were also located in the ductal plate at 17 weeks post conception (wpc) and in hepatocytes lining the scar in cirrhosis (Appendix Fig. S22). Spatial deconvolution identified LPC markers were expressed by both hepatocyte and cholangiocyte cell subpopulations (HC1/CC1) located at the scar interface in diseased liver (Appendix Fig. S21B,D,F) (So et al, 2020; Tarlow et al, 2014). Significantly, KRT23 is normally restricted to CCs, however, expression is upregulated in potential LPCs in cirrhotic liver(Guldiken et al, 2016) and was spatially expressed at the scar interface in our cirrhotic ST data (Appendix Fig. S21B,F). Similarly, the tight-junction components CLDN4 and CLDN7 are absent from HCs of healthy liver, but upregulated in hepatocytes of severely damaged human liver (Tsujiwaki et al, 2015), in agreement with our ST data (Appendix Fig. S21B,F).

Using IMC, we validated our spatial features in a patient biopsy sample of severe fibrosis (Ishak stage 3–4; Fig. 7; Appendix Fig. S23). The fibrotic scar was evident with deposition of matrix components such as collagens (COL1a and COL3), Fibronectin (FN1) and Vimentin (VIM). Although FN1 and VIM share expression with both endothelial cells and myofibroblasts, we have also previously described them as SOX9 regulated, secreted biomarkers of disease severity in progressive liver disease (Athwal et al, 2018). The myofibroblast marker, αSMA, was observed marking a layer of cells lining the scar in addition to its known localisation within the vasculature and sinusoidal space where quiescent HSCs reside. In line with our previous work (Athwal et al, 2017; Hanley et al, 2008; Pritchett et al, 2012), SOX9 was similarly detected in these αSMA rich areas as distinct elongated cells. Von Willebrand Factor (VWF) was evident in smaller vessels, distinct from αSMA, the sinusoidal space and in cells lining the matrix rich scar. Immune cell distribution was identified by the pan lymphocyte marker CD45 with high localisation within necrotic nodules and areas of αSMA positive myofibroblasts. HLA-DR was broadly associated with several immune cell types (e.g. CD11c dendritic cells, CD3 T cells and macrophages) within the scar and necrotic nodules with broad evidence of a pro-inflammatory

environment from macrophage populations located with the scar and sinusoidal space. Cholangiocytes were distinguished and highly positive for the pan cytokeratin marker (panCK), which in some cases also co-localised with CD3. However, in support of our data indicative of an altered cell state at the scar interface, cells with a hepatocyte-like phenotype (i.e. distinct shape) that were positive for panCK and localised with SOX9 had diminished expression of the epithelial transmembrane marker Syndecan 1 (SD1).

These data are reminiscent of our previous work identifying EpCAM+/SOX9+ HCs at the scar interface during progressive liver injury with similarities to ductal plate development (Athwal et al, 2017; Hanley et al, 2008). However, increasing levels of SOX9 in these cells is a prognostic and diagnostic measure that parallels severity of liver disease. As such this is more suggestive of an impaired regenerative response due to on-going liver injury.

## Parenchymal cells show dynamic cell state transitions in cirrhotic liver

Given these data, we investigated the transcriptional dynamics of parenchymal clusters using RNA velocity and trajectory inference. Overlaying RNA velocity vectors on the snRNA-seq projection uncovered temporal dynamics within parenchymal cells and remarkably, suggested a change in cell fate between HC1 and CC1, distinct from mature CCs (CC2) (Fig. 8A). Using integrated snATAC-seq data, we inferred a potential trajectory of parenchymal cell clusters (HC3 to CC2) (Fig. 8B) to highlight pseudo-temporal changes in open chromatin and gene expression associated with transitioning cell states (Fig. 8C). Although we recognised the trajectory may indeed be bi-directional, our inference of parenchymal sub-clusters identified Notch (*MAML2*), Hippo (*WWTR1, SOX9*), FGF (*FGFR2, FGFR3, FGF13*) and WNT signalling (*BICC1, ESRRG, DCDC2, ANK3*) components were highly expressed in atypical HCs (HC1) and further increased in expression along the pseudo-time from HC1 to CC1 (Fig. 8C), in agreement with our marker gene analysis (Fig. 5B,C). Moreover, these data support previous studies on signalling pathways such as Wnt, Notch and Hippo in liver regeneration, cell fate decisions and differentiation (Hu et al, 2022b; Lu et al, 2016; Okabe et al, 2016; Valle-Encinas and Dale, 2020), which when perturbed can lead to cirrhosis and HCC (Kim et al, 2017). From our integrated gene regulatory data, we identified a disease-associated parenchymal cell state (Fig. 8C,D) expressing EPCAM, consistent with our previous work defining its localisation in an aberrant SOX9 cell population at the scar-interface in progressive liver disease (Athwal et al, 2017; Athwal et al, 2018). However, our data also highlighted *ESRP1* and *ARHGEF38* as novel genes involved in the HC1-CC1 transitional axis (Fig. 8C,D). Significantly, these genes were also amongst our top differentially expressed genes (ranking 1st and 12th, respectively) identified in the HC1 subcluster (Dataset EV4).

## Integrative multi-omic analysis of parenchymal cells reveal TF dynamics underlying altered cell states in cirrhotic liver

To further interrogate the gene regulatory mechanisms underlying these altered parenchymal cell states, we identified TFs whose expression correlated with changes in motif accessibility across the pseudo-time trajectory (Fig. 9A and Dataset EV8). This approach

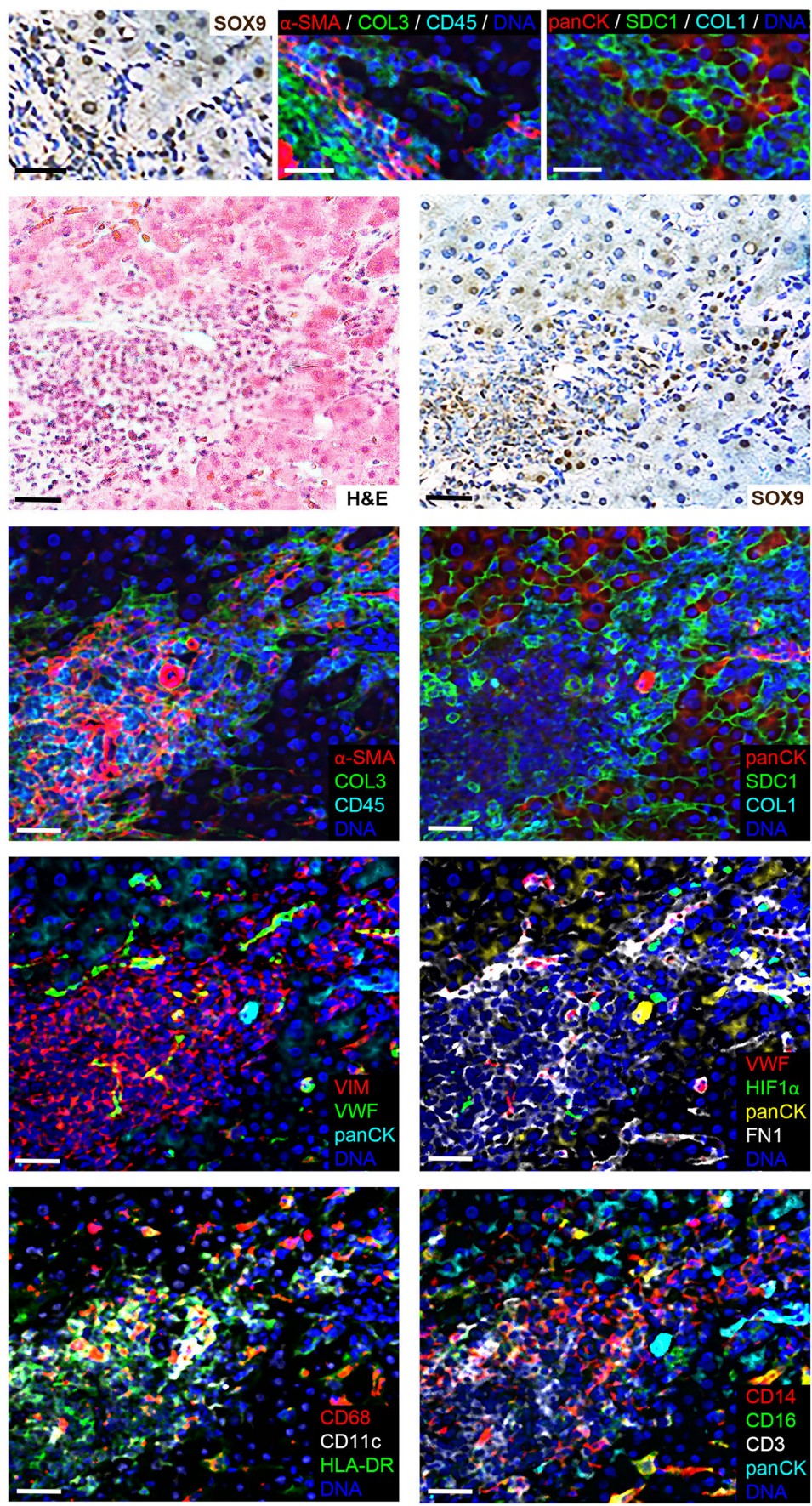

◀ **Figure 7. Characterisation and validation of spatial features in severe liver fibrosis.**

Serial sections showing imaging mass cytometry (IMC) and bright field images for SOX9 immunohistochemistry (IHC) and H&E staining as indicated (*n* = 1). Size bar = 25 μm. Source data are available online for this figure.

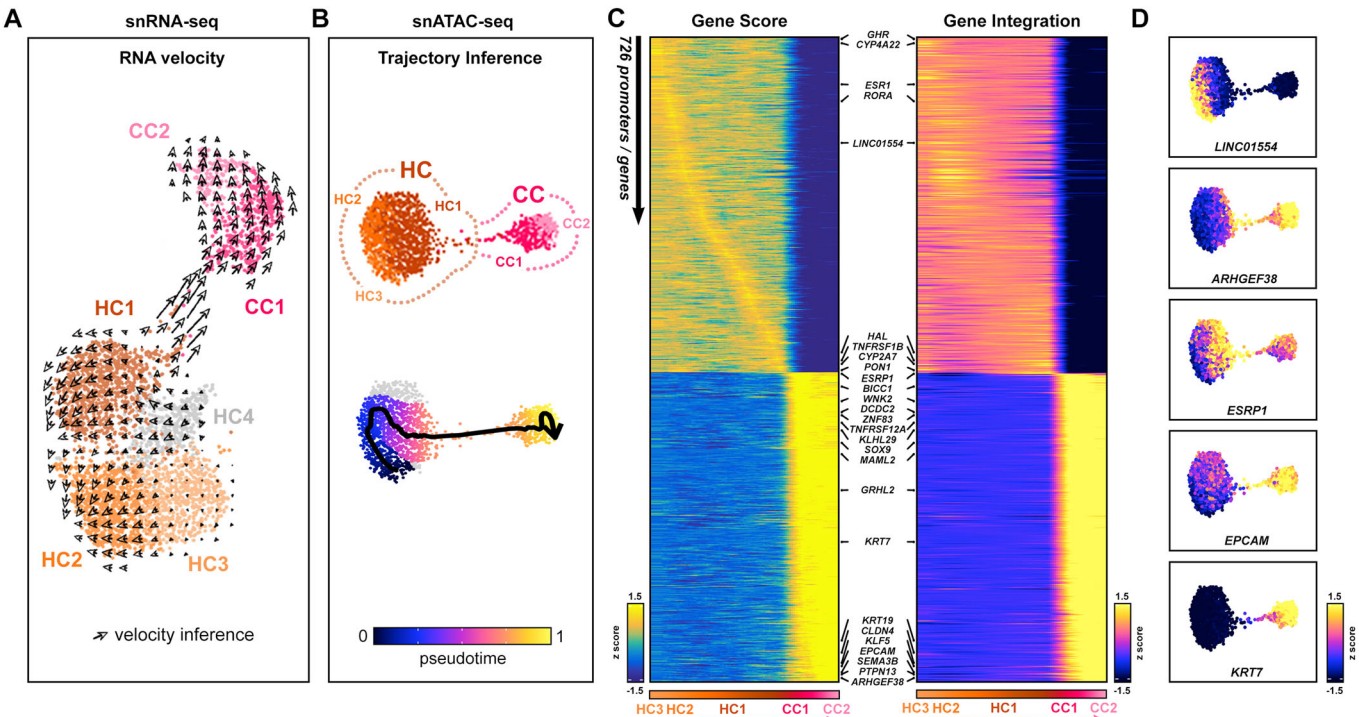

**Figure 8. Parenchymal cells display transitional cell states and directional trajectory in cirrhotic liver.**

(A, B) RNA velocity (A) and pseudo-time (B) analyses infer cell state transitions from HC1 to CC1. Each spot represents a single nuclei. (C) Integrative pseudo-time (ArchR) on snATAC-seq reveals correlated gene/enhancer accessibility (Gene Score) and expression (Gene Integration) Z-scores of significant genes involved in the transition. (D) Gene Integration plots of key transition markers showing gene expression dynamics (Z-score) along the pseudo-time. Data information: Hepatocytes (HC), cholangiocytes (CC).

revealed the dynamics of several core hepatic TFs (*HNF4A*, *NR5A2* (*LRH-1*), *HNF1B*, *ONECUT1*, *GATA6*, *KLF5*) associated with cell states and their potential dysregulation due to ongoing liver disease. The transition of hepatocytes (HC2/HC3) to a disease-associated state (HC1) correlated with the downregulation of several orphan nuclear receptors (*HNF4A*, *NR1I2/I3*, *NR2C1*, *RORA/C*) (Fig. 9A,B). Indeed, HNF4A is a master regulator of hepatocyte cell identity and several studies have shown its impairment in human chronic liver disease and potential as a therapeutic target in liver cirrhosis (Argemi and Bataller, 2019; Berasain et al, 2003; Guzman-Lepe et al, 2018; Joo et al, 2019; Yang et al, 2021).

Our analysis also revealed the transient expression and increased chromatin accessibility of several TFs in HC1 (*TCF7L1*, *NR5A2* (*LRH-1*), *FOXO3*) just prior to the transition to CC1 (Fig. 9A). Interestingly, these TFs are implicated in Wnt signalling. TCF7L1 modulates Wnt target gene expression in the presence/absence of β-catenin, while *NR5A2* (LRH-1) itself is a β-catenin target gene with fundamental roles in fetal liver development and adult hepatocyte cell identity (Joo et al, 2019). Conversely, FOXO3 negatively regulates β-catenin (*CTNNB1*) in hepatocellular carcinoma and supresses hepatocyte proliferation during liver regeneration (Liang

et al, 2022; Yang et al, 2016). Collectively, our data suggests these TFs are transiently upregulated in disease-associated cell states, and may play a role in cell fate and disease progression during progressive liver fibrosis (Boulter et al, 2012).

Consistent with our previous work, integrative TF analysis identified the downstream effector, *SOX9* across transitional cell states. Our data showed expression and enhancer accessibility in CC1, suggesting transition toward a biliary cell fate (Fig. 9A, B). In addition, we also identified *ONECUT1, SOX13, RFX7* and *ZNF148* as transitional TFs. In particular we noted *ONECUT1* expression and motif accessibility spanned the transition from HC1 to CC1 (Fig. 9A,B). During liver development, both SOX9 and ONECUT1 are involved in ductal plate specification. However, although SOX9 expression is maintained in mature biliary cells into adult, HNF6 is lost from the biliary lineage and expressed in SOX9 negative hepatocytes (Athwal et al, 2017; Hanley et al, 2008; Limaye et al, 2008). Given these data and gene expression profiles, we investigated how SOX9 and ONECUT1 may regulate the transitional landscape of diseased states within the HC1-CC1 axis. Consistent with their roles in ductal plate during liver development, our GRN identified SOX9/ONECUT1 co-regulation of genes

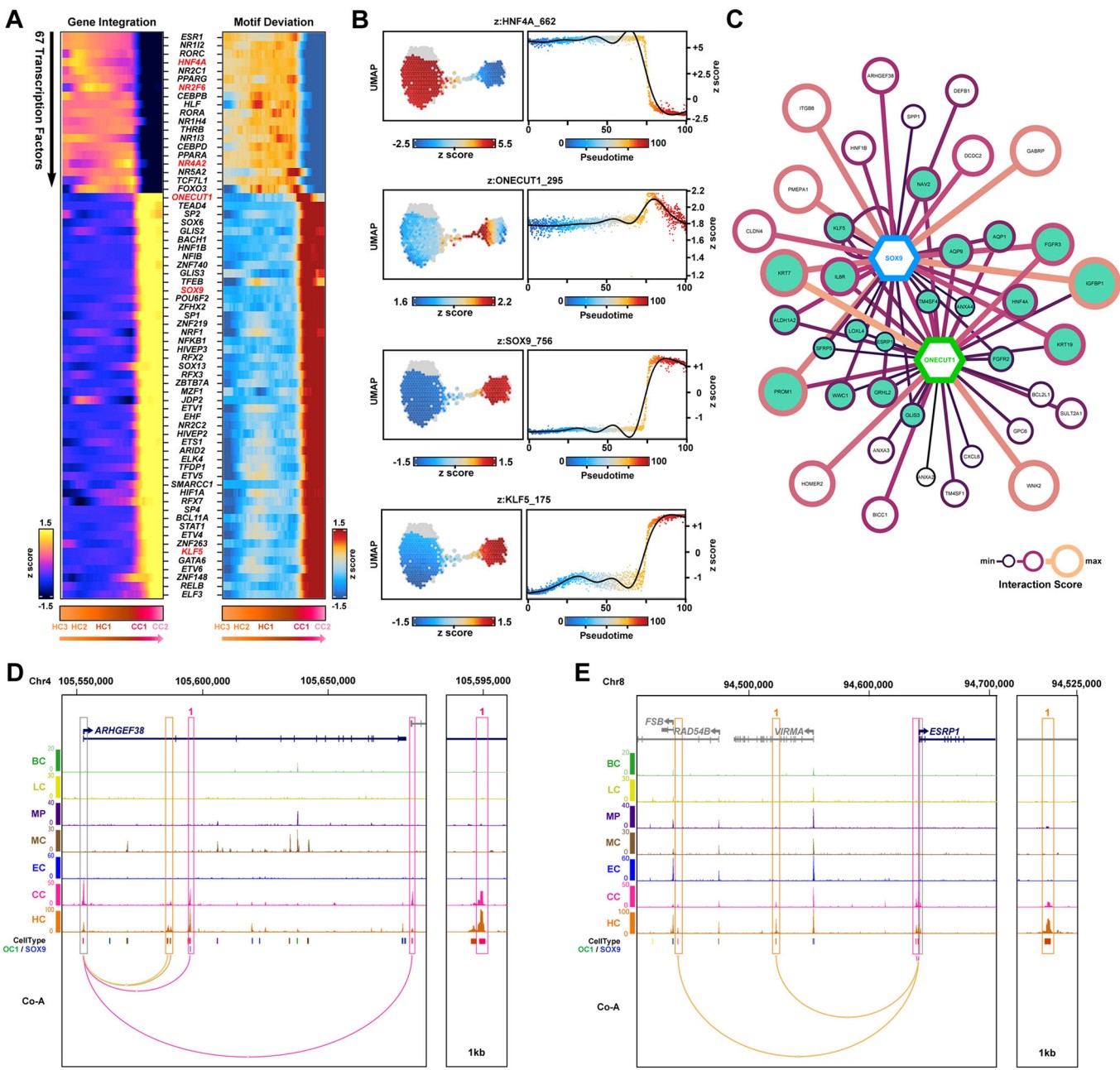

**Figure 9. Integrative pseudo-time reveals TF dynamics underlying disease-associated parenchymal cell states at scar interface.**

(A, B) Integrated multi-omic analyses (ArchR) of parenchymal cell subpopulations (HC3-CC2) reveal the correlated expression of TFs and enhancer accessibility (motif deviation) Z-scores along the transition from HCs to CCs. Select TFs of interest are highlighted (red). (C) Gene regulatory network (GRN) visualisation (Cytoscape) for select targets of SOX9 (blue) and ONECUT1 (green), constructed using enhancer-gene co-accessible peak2gene interactions with corresponding TF motif. Potential co-regulated targets (teal). (D, E) UCSC browser screenshots of cell-type snATAC-seq tracks, show cell-type co-accessible (Co-A) enhancer-gene interactions (HC, orange; CC, pink) and TF motif locations (SOX9, blue; OC1; green) for *ARHGEF38* (D) and *ESRP1* (E). Data information: In (B), UMAP motif accessibility plots (z-score) of select TFs within parenchymal cells. Hepatocytes (HC), endothelial cells (EC), cholangiocytes (CC), macrophages (MP), mesenchymal cells (MC), T lymphocytes (TC), B lymphocytes (BC).

associated with a biliary and LPC phenotype (including *KRT19*, *KRT7*, *PROM1*, *FGFR2*, *TM4SF4* and *ANXA4*; Fig. 9C and Dataset EV8) (Segal et al, 2019). Genes involved in signalling pathways typical of cell fate transitions in hepatocytes and cholangiocytes were also highlighted (*SFRP5*, *FGFR2*, *FGFR3* and *GRHL2*) (Fig. 9C) (Segal et al, 2019). Drawing on novel genes identified in the HC1-

CC1 transitional axis, our snATAC-seq data revealed co-accessible interactions for *ARHGEF38* enriched for SOX9 motifs (Fig. 9C,D), whereas *ESRP1* was co-accessible with both SOX9 and ONECUT1 identified enhancers (Fig. 9C,E). Although promoter activity was evident in both the HC and CC population, our snATAC-seq data contained an overabundance of transitional cell-types from HC1

and CC1, respectively (Figs. 2F and 9B,D,E), with 27% of enhancers co-accessible across both HC and CC cell types (Appendix Fig. S6B). Moreover, peak enhancer accessibility for both *ESRP1* and *ARHGEF38* was more prevalent in the HC population, largely due to HC1 overabundance (Fig. 9D,E). As validation, we localised SOX9 and ARHGEF38 in hepatocyte-like cells lining the scar in a mouse model of liver fibrosis due to carbon tetrachloride (CCl$_4$) injury (Appendix Fig. S24A). Whereas, mechanistically, and consistent with our GRN discoveries, overexpression of SOX9 was associated with driving *ARHGEF38* expression in HEPG2 cells, and ONECUT1 was associated with increased *ESRP1* expression (Appendix Fig. S24B,C).

These data provide novel insight into liver-specific mechanisms driving disease. In particular, the transcriptional diversity driven by alternative splicing events during disease-associated cell transitions is not well understood. Significantly, ESRP1 is an epithelial cell-specific RNA-binding protein that regulates alternative splicing of several genes, including *FGFR2* highlighted by our GRN and others involved in epithelial-mesenchymal transition (EMT) (Bebee et al, 2015; Lee et al, 2022). As such, it has received much attention playing a role in tumour motility and invasiveness. Similarly, ARHGEF38, a RhoGEF implicated in tumorigenesis and metastasis (Liu et al, 2020), is associated with cancer progression in prostate. However, these genes have more divergent roles relevant to fibrosis and potentially demarcate early transitional events in progressive disease. In particular, gene splicing is a critical aspect of gene expression driving human development and cell differentiation (Lee et al, 2022), it seems conceivable that expression of ESRP1 in the HC1-CC1 cell populations may recapitulate fetal gene splicing to create a signature indicative of attempted regeneration. Moreover, as a direct consequence of fibrosis, mechanical properties of ECM also regulate alternative splicing, including those driven by ESRP1 (Wang et al, 2023), through effects on intracellular contractility involving RhoGEFs. In combination, this microenvironment and TF profile would potentially alter the cell state and behaviour of our spatially identified HC1-CC1 population located at the scar interface in liver disease.

# Discussion

Single-cell sequencing technologies have advanced our understanding of cellular heterogeneity in both normal and diseased human liver (Aizarani et al, 2019; Dobie et al, 2019; MacParland et al, 2018; Matchett et al, 2024; Planas-Paz et al, 2019; Ramachandran et al, 2019). More recently ST studies have provided insight into the distribution of specific cell-types in healthy and diseased states, zonation in normal liver and hepatic macrophage identity in healthy and obese liver (Chung et al, 2022; Guilliams et al, 2022; Hildebrandt et al, 2021; Hu et al, 2022a). Here, we combine spatially resolved and single nucleus gene expression with chromatin accessibility to provide a comprehensive map in tissue from patients with liver cirrhosis and identify changing cell states associated with the fibrotic scar during disease.

Our computational approaches have identified a series of spatially resolved maps to characterise changing cell states as part of the fibrotic niche in diseased liver. In keeping with their role in excessive matrix deposition, we identified specific gene signatures reminiscent of activated HSCs (aHSCs; liver-specific myofibroblasts) located within the scar. Significantly these analyses also identified scar-associated expression of novel aHSC genes, *QSOX1* and *RAMP1*. QSOX1 is

essential for incorporating laminin into the ECM, resulting in an aHSC response through integrin-mediated cell adhesion, migration and signalling (Coppock et al, 2000; Ilani et al, 2013). Aberrant expression of laminin and QSOX1 is also observed in various cancers (Finak et al, 2008; Soloviev et al, 2013; Sung et al, 2018) while inhibition of QSOX1 is associated with fewer myofibroblasts, a disorganised ECM and decreased matrix stiffness in the tumour microenvironment (Feldman et al, 2020). RAMP1 is a receptor for the neuropeptide calcitonin gene-related peptide (CGRP) and functionally links sensory innervation with liver regeneration through YAP/TAZ regulation and immune infiltration (Holzmann, 2013; Laschinger et al, 2020). Thus, targeting these factors may provide a novel therapeutic route to limit progressive fibrosis and improve regenerative response.

Similar to aHSCs, several immune cell populations were also spatially resolved within the fibrotic scar, including resident macrophages, inflammatory macrophages and lymphocyte populations. Significantly, the localisation and gene signatures associated with inflammatory macrophages were indicative of an injury response suggestive of on-going damage/repair of surrounding cells. In agreement with previous scRNA-seq studies (Aizarani et al, 2019; Dobie et al, 2019; MacParland et al, 2018; Planas-Paz et al, 2019; Ramachandran et al, 2019), we also spatially located a subset of scar-associated endothelial cells that were expanded and localised to regions of scarring in liver cirrhosis. These data further emphasise the multicellular response to the local environment in liver disease.

Our multi-omic approach combining spatial and single cell data provided an opportunity to investigate how hepatocyte cell states are influenced by their tissue environment. Our results highlighted a spatial pattern of gene modules specifically localised at the scar interface. Unbiased computational analysis identified these sub-populations containing genes characteristic of both hepatocytes and cholangiocytes (HC1/CC1). Previous studies aimed at understanding liver regeneration suggest that, depending on the injury, both cholangiocytes and hepatocytes are capable of functioning as facultative stem cells and transdifferentiate into either cell-type. However, the majority of these studies have been inferred from specific mouse models of severe injury (Ding et al, 2010; Espanol-Suner et al, 2012; Gilgenkrantz and Collin de l'Hortet, 2018; Raven et al, 2017). In this study, our data provided an opportunity to investigate the regenerative mechanisms and transitional states of hepatocytes and cholangiocytes directly in human. Through pseudo temporal analysis of our single nuclei data, we uncovered a directional trajectory and transitional state between hepatocytes and cholangiocytes. Mechanistically, our gene expression and transcriptional motif analysis pointed toward signalling in cirrhosis involving crosstalk between Hippo, Notch and Wnt; all perturbed and associated with liver disease (Kim et al, 2017).

Significantly, we have previously characterised hepatocytes lining the scar in models of human and mouse liver disease and shown an impaired response with cells expressing both hepatocyte (e.g. α1AT) and cholangiocyte (CK19) markers. Moreover, as an ectopic and impaired regenerative response we have also identified these cells as EPCAM/SOX9+ (similar to ductal plate cells during human liver development) (Athwal et al, 2017). In patients with liver disease, the extent of ectopic SOX9 expressing cells in liver biopsies during early phases of liver fibrosis predicts progressive disease within 3 years (Athwal et al, 2017). As further insight into the mechanisms underpinning these EPCAM / SOX9+ altered cell states at the scar interface, we identified ONECUT1 as a critical

transcriptional regulator of disease associated hepatocytes. Analysis of the integrated snRNA and ATAC-seq data identified a novel transitional axis, marked by *ESRP1* and *ARHGEF38* expression, driven by ONECUT1 and SOX9 through a co-accessibility enhancer network regulating these transitional gene states in cirrhosis. These data also highlight previously unexplored mechanisms involving ESRP1 and gene-splicing events involved in transcriptional reprogramming in progressive liver disease. These may be linked to alternative promoter use and splicing of *HNF4A* during disease, in response to both TGFβ signalling and the mechanical properties of the scar (Argemi et al, 2019; Wang et al, 2023). Of interest, the hepatocyte gene, *HNF4A*, has two promoters and encodes at least nine isoforms through differential splicing (Harries et al, 2008). In alcoholic hepatitis, the core profibrotic cytokine TGFβ induces fetal *HNF4A-P2* promoter use rather than adult *HNF4A-P1*, resulting in dysregulated gene expression including an EMT-type response and increased levels of GRHL2 (Argemi et al, 2019).

Collectively, we have provided a publicly available map and resource for further integrative studies requiring insight from single cell transcriptomics, epigenomics and spatial gene expression in human liver (https://cellxgene.bmh.manchester.ac.uk/liver_disease/). In recognition of the limitations relating to patient samples used to uncover potential molecular pathways, our data provides a framework for further integration with larger patient orientated studies that will improve mechanistic insight into liver disease. In the context of the underlying disease (metabolic associated steatotic liver disease; MASLD), the gene regulatory networks may provide mechanistic insight to identify the causal genes associated with unmapped genome-wide variants and disease risk. Overall, these data will facilitate studies aimed at identifying new markers for early detection of liver disease, therapeutic targets to improve liver function and advance our understanding of its regenerative capacity directly during progressive disease.

# Methods

### Reagents and tools table

| Reagent/Resource | Reference or Source | Identifier or Catalog Number |
|---|---|---|
| **Experimental models** | | |
| HepG2 | ATCC | HB-8065 |
| C57BL/6 | Charles River | N/A |
| **Recombinant DNA** | | |
| pcDNA3.1 | ThermoFisher | V79020 |
| pcDNA3.1-hSOX9 | https://doi.org/10.1002/hep.25758 | N/A |
| pFR_HNF6/ONECUT1 | Addgene | 31099 |
| **Antibodies** | | |
| anti-ARHGEF38 | Invitrogen | PA5-57695 |
| anti-SOX9 | Merck | ab5535 |
| anti-TROP2 | Abcam | ab214488 |
| Biotinylated Anti-Rabbit Secondary | Vector Laboratories | BA-1100 |
| Strep-HRP | Vector Laboratories | SA-5004 |

| Reagent/Resource | Reference or Source | Identifier or Catalog Number |
|---|---|---|
| Antibodies for IMC | This study | Appendix Table S10 |
| **Oligonucleotides and other sequence-based reagents** | | |
| Dual Index Plate TS set A | 10X Genomics | 1000251 |
| Dual Index Kit TT Set A | 10X Genomics | 1000215 |
| qPCR primers | IDT Technologies | Sequences in methods |
| **Chemicals, Enzymes and other reagents** | | |
| Visium FFPE Kit | 10X Genomics | 1000336 |
| Visium Frozen Kit | 10X Genomics | 1000187 |
| Visium Tissue Optimisation Slide and Reagent Kit | 10X Genomics | 1000193 |
| Chromium Next GEM Single Cell ATAC kit | 10X Genomics | 1000406 |
| Chromium Next Gem Single Cell 3′ Kit | 10X Genomics | 1000269 |
| Carbon Tetrachloride | Merck | 289116 |
| Olive Oil | Merck | O1514 |
| RNAse inhibitor | Merck | 3335399001 |
| DAB Substrate Kit | ThermoFisher Scientific | 34002 |
| Maxpar® X8 Antibody Labeling Kit | Standard Biotools | 201141A |
| Ir191/193 DNA intercalator | Standard Biotools | 201192A |
| Lipofectamine 3000 | Invitrogen | L3000008 |
| RNEasy RNA extraction Kit | Qiagen | 74104 |
| High-capacity RNA-to-cDNA kit | Applied Biosciences | 4368814 |
| **Software** | | |
| BBrowser v3.1.7 | https://bioturing.com/bbrowser | N/A |
| Loupe Browser 6.2.0 | https://www.10xgenomics.com/ | N/A |
| Adobe Photoshop CS6 | https://www.adobe.com | N/A |
| cell2location v0.04-alpha | https://doi.org/10.1038/s41587-021-01139-4 | N/A |
| Cellxgene v1.1.1 | https://doi.org/10.1101/2021.04.05.438318 | N/A |
| UCSC Genome Browser hg38 | https://genome.ucsc.edu/ | N/A |
| MEMEsuite: FIMO v5.5.7 | https://meme-suite.org/meme/tools/fimo | N/A |
| Cytoscape v3.9.1 | https://cytoscape.org/ https://doi.org/10.1101/gr.1239303 | N/A |
| Enrichr | https://maayanlab.cloud/Enrichr/ https://doi.org/10.1186/1471-2105-14-128 | N/A |
| Cell Ranger v3.1.0 | https://www.10xgenomics.com/ | N/A |
| Space Ranger | https://www.10xgenomics.com/ | N/A |

| Reagent/Resource | Reference or Source | Identifier or Catalog Number |
|---|---|---|
| Scanpy | https://doi.org/10.1186/s13059-017-1382-0 | N/A |
| ArchR | https://doi.org/10.1038/s41588-021-00790-6 | N/A |
| scVelo v0.2.4 | https://doi.org/10.1038/s41587-020-0591-3 | N/A |
| Harmony | https://doi.org/10.1038/s41592-019-0619-0 | N/A |
| Macs2 | https://doi.org/10.1186/gb-2008-9-9-r137 | N/A |
| CIS-BP | https://cisbp.ccbr.utoronto.ca/ https://doi.org/10.1016/j.cell.2014.08.009 | N/A |
| CyTOF | Standard Biotools | N/A |
| MCDViewer | Standard BioTools | N/A |
| ImageJ | https://imagej.nih.gov/ij/index.html | N/A |
| GraphPad Prism 10 | GraphPad | N/A |
| Other | | |

## Human tissue

Informed consent was obtained from all human subjects and experiments conformed to the principles set out in the WMA Declaration of Helsinki and the Department of Health and Human Services Belmont Report.

Liver tissue was obtained following ethical approval and informed consent (National Research Ethics Service, REC14/NW1260/22) from the Manchester Foundation Trust Biobank. Fresh, unfixed tissue was obtained intraoperatively from patients undergoing surgical liver resections at the Manchester Royal Infirmary. Tissue for the experimental cohort was collected from three independent patients. All had been diagnosed with early cirrhosis (Childs-Pugh A), confirmed histologically by an expert clinical histopathologist in the non-malignant regions of the resection as part of the clinical care pathway at Manchester Royal Infirmary. Additional patient details can be found in Table EV1. Tissue was snap frozen within 30 min of collection and stored at −80 °C. For the validation cohort biopsy tissue for spatial transcriptomics was collected following ethical approval and informed consent (REC20/NS/0055) as part of the Innovate funded project on Integrative Diagnostics for the early detection of liver disease (ID-LIVER). All biopsies are graded for fibrosis as part of the clinical care pathway at Manchester Royal Infirmary by an expert clinical pathologist. Sample biopsy details can be found in Table EV1.

## Mouse tissue

Animals were housed and maintained on a standard 12-h light/dark cycle and provided with food and water ad libitum. Animal work was undertaken under relevant UK Home Office Project and Personal Licenses and with approval of the University of Manchester Animal Welfare Ethical Review Board. Eight-week-old mice on the C57BL/6 background were injected twice weekly with 2 μl/g of 25% $CCl_4$ in olive oil for 16 weeks. Mice were sacrificed 24 h after the final injection. All randomly assigned experimental and controls were littermate sex- and age-matched mice and analysed blind to the treatment. Liver tissue was fixed overnight in 10% NBF before transfer into 70% ethanol and processing into FFPE tissue blocks and sectioned as previously described (Athwal et al, 2017; Jokl et al, 2023; Martin et al, 2016).

## Sample processing, library preparation and sequencing

Snap frozen liver tissue was rapidly thawed in a 37 °C water bath for ~60 s. Tissue was minced on ice with sterile razor blades alongside addition of cold lysis buffer (10 mM Tris HCl, 10 mM NaCl, 3 mM $MgCl_2$, 0.1% IGEPAL, 0.1% Tween20, 0.01% Digitonin, 0.2 U/μl RNAse inhibitor). Minced tissue was transferred to a falcon tube and incubated on ice in lysis buffer for 1 h with periodic gentle vortexing every 10–15 min. Tissue was passed through 40 μm and 20 μm filters and centrifuged at $500 \times g$ for 10 min at 4 °C. The pellet was gently resuspended in wash buffer (10 mM Tris HCl, 10 mM NaCl, 3 mM $MgCl_2$, 0.1% Tween20, 0.2 U/μl RNAse inhibitor). The suspension was split in half to provide nuclei for independent snATAC-seq and snRNA-seq runs. The centrifugation and wash step was repeated before a final centrifugation at $500 \times g$ for 5 min at 4 °C. The pellets were then resuspended in either 0.1% ultrapure BSA in PBS plus 0.2 U/μl RNAse inhibitor for snRNA-seq, or 1x nuclei buffer diluted from 20x nuclei buffer (10x Genomics) for snATAC-seq.

Nuclei were processed through Chromium Next GEM Single Cell ATAC and Chromium Next GEM Single Cell 3′ protocols by the University of Manchester Genomic Technologies Core Facility. Briefly, for snATAC-seq nuclei suspensions were incubated in a Transposition Mix to fragment accessible DNA and ligate adaptor sequences to the fragments prior to GEM generation and barcoding, cleanup and library construction. For snRNA-seq, GEMs were generated for barcoding and reverse transcription. cDNA was amplified and a library generated after fragmentation, end repair and A-tailing and adaptor ligation steps. Sequencing was carried out on the Illumina NextSeq 500 Platform.

## snRNA-seq data processing and clustering

### Data pre-processing

The sequence files generated from the sequencer were processed using 10x Genomics custom pipeline Cell Ranger v3.1.0 (Zheng et al, 2017). The fastq files generated from the pipeline are then aligned to the hg38 custom genome with all the default parameters of Cell Ranger. Cell Ranger identifies the valid barcodes with nuclei and counted UMIs associated to each nucleus. It uses STAR aligner to align reads to genome while discarding all the counts mapping to multiple loci during counting. The uniquely mapped UMI counts are reported in a gene by cell count matrix represented as sparse matrix format. Cell Ranger outputs 9754 nuclei which is then taken for further downstream QC.

### Filtering

The low-quality cells were removed from the dataset, to ensure that the technical noise does not affect the downstream analysis. We

looked into three commonly used parameters for cell quality evaluation, the number of UMIs per cell barcode (library size), the number of genes per cell barcode and the proportion of UMIs that are mapped to mitochondrial genes (Bourgon et al, 2010). Cells that have lower UMI counts than 0.5 Median Absolute Deviation (MAD) for these three parameters were filtered out. To check whether there are cells that have outlier distributions, which can indicate doublets or multiplets of cells, violin plots were used on these three parameters. The outlier cells that have total read counts more than 100,000 were removed, as potential doublets. The cutoff for mitochondrial reads was 17.45%. After filtering, 6170 nuclei remained for downstream analysis.

### Classification of cell-cycle phase

The cyclone method was used to calculate for each cell the score of G1 phase and G2M phase (Scialdone et al, 2015). Cyclone uses a pre-build model that is trained on a known cell-cycle dataset where the sign of difference in the expression between a pair of genes was used to mark the cell-cycle stages. The cell-cycle phase of each cell is identified based on the observed sign for each marker pair of each of the phases.

### Normalization

To make sure that the counts are comparable among cells, we applied library-size normalization, where each cell is normalized by the total counts over all the genes in the dataset. The total counts were set to 10,000 reads per cell. Logarithmic transformation (natural log) was then applied to the data matrix.

### Visualization & clustering

As a first step for visualization and clustering the highly variable genes (HVG) were identified using Scanpy's *highly_variable_genes()* function (Wolf et al, 2018). This function identifies HVGs by implementing a dispersion-based method. In this approach the normalized dispersion is calculated by first grouping the genes in multiple different bins and then scaling with the mean and standard deviation of the gene dispersion for which the mean expression falls into a given bin. These HVG genes were then used to reduce the dimensions of the dataset using PCA. The dimension of the dataset was further reduced to 2D using t-SNE and UMAP with 40 PCA components. To calculate the UMAP, a neighbourhood graph was first generated of cells using Scanpy's *neighbors()* function with *the n_neighbors* parameter set to 10. Then, the UMAP was generated using this neighbourhood graph. For clustering, the SNN (Shared Neighbourhood Network) method was used, which identified 7 clusters in the dataset.

### Identification of marker genes

To identify marker genes for each cluster, Scanpy's *rank_gene_groups()* function with the Wilcoxon method was used. Differentially expressed genes are reported based on a comparison of each cluster's marker genes against all other clusters. These marker genes were then used to annotate the cell types of a cluster.

### RNA-velocity

scvelo v0.2.4 was used to calculate the RNA-velocity of the data (Bergen et al, 2020). The spliced vs un-spliced ratio was calculated using the velocyto CLI method. This method gives a ratio of 18% spliced vs 82% un-spliced for this data. The default stochastic method to calculate the velocity was used.

### Integration of healthy and cirrhotic snRNA-seq

To compare our cirrhotic snRNA-seq data with healthy liver, we downloaded raw counts and associated metadata of three healthy samples from GSE185477 (C41_TST, C70_TST, C72_TST) (Andrews et al, 2022). We merged the cells into one object using Scanpy, filtered cells according to the published metadata and integrated both datasets (healthy, cirrhotic) using Harmony (Korsunsky et al, 2019). This algorithm enables cells from different technologies, batches and conditions to be projected into a shared embedding where cells are grouped by cell-type rather than dataset conditions, allowing true biological differences from rare cell populations to be uncovered (Korsunsky et al, 2019). We used the latent dimension from Harmony to calculate the UMAP, together with previously annotated cell-type metadata from both datasets, to compare gene expression between cell sub-populations across conditions.

### Data pre-processing

snATAC-seq fastq files were processed using 10X Genomics custom pipeline for snATAC-seq, cellranger-atac v.1.2.0 and was mapped to hg38 custom genome with all the default parameters. The pipeline identified 4743 nuclei with 6238 median fragments per nucleus. With about 42.5% of fragments overlapped called peaks.

### Cell quality control

To conduct quality control of snATAC-seq data, two parameters were considered, the number of unique nuclear fragments and the signal-to-backround ratio and the signal-to-background ratio. The former is represented as the fragments that are not mapping to mitochondrial DNA, while the latter, is measured as TSS enrichment score, which is calculated as the ratio between the peak enrichment at transcription start site raltive to the flaning region. Cells having lower than 1000 ATAC-seq fragments or lower than 4 TSS enrichment are filtered out. 4680 cells passed this filtering for downstream analysis.

Potential doublets were identified using ArchR's *addDoubletScores()* function (Granja et al, 2021). This function first synthesizes in-silico doublets and then cells that are within these synthetic doublets's neighbourhood in the UMAP embedding are identified as potential doublets. This procedure is iterated for thousands of times to calculate the confidence on a cell being a potential doublet. To identify these doublets, the default parameters were used, where 219 of 4680 cells (4.7%) were identified as potential doublets and were filtered out.

### Dimensionality reduction and clustering

The iterative Latent Semantic Indexing (LSI) was applied on a genome-wide 500-bp sized tiles (Granja et al, 2021). Initially the iterative LSI identifies lower resolution clusters that mostly correspond to major cell-types. Then, the average accessiblity of each peak across these clusters is computed and being used in order to identify peaks that are most variable across the dataset. This information is used in the subsequent iteration for the identification of the informative clusters. To cluster the cells, ArchR's *addClusters()* function was used, which is based on a graph clustering implemented in Seurat (Stuart et al, 2019), where clusters are identified in LSI sub-space. To visualize our snATAC-seq data, a 2D UMAP embedding generated with ArchR's *addUMAP()* function was used.

### Cell annotation and peak calling

Cells from snATAC-seq were annotated by integration with our snRNA-seq data. Cells from snATAC-seq were aligned to cells from snRNA-seq by comparing gene expression data from snRNA-seq with the gene score data from snATAC-seq. The gene score was calculated by counting the accessiblity within the gene body and 100 KB upstream/downstream of the Transcription Start Site (TSS), weighted based on the distance from the TSS. Cell type labels from the aligned snRNA-seq are then transferred to the corresponding snATAC-seq cells. The gene expression from the aligned snRNA-seq are also assigned to snATAC-seq data as a pseudo-gene expression value.

As the snATAC-seq data are essentially binary, peaks can not be called on individual single-cells. Therefore, pseudo-bulk replicates were defined based on clusters and called peaks on these pseudo-bulk data. Macs2 was utilised for peak calling and, for the downstream analysis, a 501 bp fixed width peak was used (Weirauch et al, 2014). To identify marker peaks for each cluster, the *getMarkerFeatures()* function was used, which compares each cell group to its background. These marker peaks are then used to conduct motif enrichment anlaylsis for each cell-type. For motif annotations, CIS-BP (Catalog of Inferred Sequence Binding Preferences) database was used (Weirauch et al, 2014). Peaks that are linked to genes, were identified by using the *addPeak2GeneLinks()* function. To infer these links, this function looks at the correlation between the peak accessibility and the gene expression, which gives the potential regulatory regions for each peak.

### Trajectory analysis

To order cells in a pseudo-time, ArchR aligns cells in N-dimensional LSI subspace based on a user-defined trajectory backbone. The mean cordinates for each cluster in N-dimensional suspace are then calculated and only the cells for which the Euclidean distance between these mean cordinates are within the top 5% of all cells are retained. The cells are then ordered based on the distance from the cell's own cluster and its next cluster mention in the backbone. These cells are then aligned along the embedding. Finally, ArchR scales the alignment to 100. The changes in GeneScore and GeneIntegration matrix across pseudo-time were visualized using heatmap plots based on *plotTrajectoryHeatmap()* function. To have integrated pseudo-time heatmaps, the *correlateTrajectories()* function was used, to identify features that changes in a correlated manner in two different matrices.

## Gene regulatory network inference

Using snATAC-seq data, all significant (correlation >0.45) co-accessible peak2gene interactions were identified and enhancers were annotated to cell-type clusters. From this library, we extracted all parenchymal specific (HC/CC) interactions associated with genes in the HC-CC pseudo-time transition and scanned enhancer peaks for SOX9 (MA0077.1) and ONECUT1 (MA0679.1) motifs using FIMO (Grant et al, 2011). Utilizing this catalog of potential SOX9 and/or ONECUT1 regulated targets, we constructed GRNs using Cytoscape v3.9.1 (Shannon et al, 2003) for up to 40 genes of interest, visualizing node size/colour and edge thickness/colour based on interaction significance score (-LOG10(FDR)).

## Visium ST sample preparation, optimisation, library preparation and sequencing

### Experimental cohort

Snap frozen cirrhotic liver resections (samples a–c) were embedded in optimal cutting temperature (OCT), frozen and cryosectioned with 10 μm thickness at −10 °C (Leica CM1900) with a chamber temperature of −20 °C. Sections were transferred onto capture areas of a chilled Visium Tissue Optimisation Slide and stored for 24 h at −80 °C before continuing with the manufacturer protocol the following day. Tissue sections were permeabilized, captured mRNA was reverse transcribed using fluorescently labelled oligonucleotides and imaged in situ using a high-resolution slide scanner with TRITC filters (3D-Histech Pannoramic-250). A tissue permeabilization time-series determined 12 min was the optimum permeabilization duration for human liver tissue (Appendix Fig. S2).

### Validation cohort

FFPE embedded core liver biopsies from clinically diagnosed fibrotic patients without HCC (samples d–g) were sectioned at 5 μm and mounted on a Visium FFPE Gene Expression slide before following the standard Visium FFPE protocol.

Visium Spatial Gene Expression slides for fresh-frozen (samples a-c) and FFPE (samples d–g) were prepared according to the manufacturer protocols and stained for hematoxylin and eosin (H&E) prior to imaging on a high-resolution slide scanner (3D-Histech Pannoramic-250). Spatial libraries were constructed, QC'd, and sequenced at depth on NextSeq500 platform (Illumina). Following sequencing, spatial reads were assigned back to the corresponding tissue section using Space Ranger software and the data explored using Loupe software and other bioinformatic tools. All experimental samples (samples a-c) were sequenced to a mean depth of 255.3 M reads, detecting on average 2986 unique genes and 11,995 UMIs per spot, with the entire dataset comprising a total of 8028 spots across four tissue sections, including one technical replicate (Appendix Fig. S3). Validation cohort samples (samples d–g) were sequenced to a mean depth of 79.4 M reads, detecting on average 3339 unique genes and 11,417 unique UMIs per spot, with the entire dataset comprising a total of 2080 spots across four tissue sections (Appendix Fig. S8).

## Visium ST data processing and clustering

Space Ranger analysis pipelines were used to process Visium Gene Expression data. For each sample, the fiducial frames from H&E images were manually aligned within Space Ranger, spots over tissue selected and processed using the standard fresh-frozen (samples a–c) or FFPE (samples d–g) sample workflows. UMI counts were handled by Space Ranger, prefiltering the data to remove duplicates and sequencing errors in UMIs. Regions with unusually low UMIs due to sectioning/processing artefacts were reviewed after the workflow and if deemed necessary, the sample re-processed omitting these spots.

Dimensionality reduction was handled by Space Ranger, using Principle Component Analysis (PCA). The data was visualised in 2D by passing PCA-reduced data into t-Stochastic Neighbour Embedding (t-SNE) (Maaten, 2014) and Uniform Manifold

Approximation and Projection (UMAP) (Becht et al, 2018) algorithms, both plotted and visualised within Loupe.

Space Ranger performed Graph-based (Louvain) (Blondel et al, 2008) and K-means clustering (across a range of values) to group spots by expression similarity. For each clustering method, differentially expressed genes (DEGs) per cluster were identified relative to the rest of the sample using the Loupe browser. Gene ontology (GO) analysis was performed using Enrichr (Chen et al, 2013) on the top 50 DEGs per cluster. GO data was presented as a Manhattan-style dot plot and significant GO terms were selected per cluster. GO terms were mapped spatially by plotting the average expression signature (Log2 average UMIs) of 5 significantly enriched genes per term. Spatial gene expression was visualised within Loupe browser. For each gene, violin plots were used to set upper and lower UMI thresholds before visualising.

## Spatial cell-type deconvolution and co-location analysis

To map the spatial distribution of individual cell-types and infer cell-type proportions for each spatial spot, we used cell2location ('v0.04-alpha') (Kleshchevnikov et al, 2022); a python tool based on Bayesian modelling. This tool was selected in preference to other methods due to its high accuracy and computational efficiency. Cell2location estimates relative and absolute cell abundance at each spatial spot by decomposing the gene expression counts in ST data into a set of reference signatures (cell-types). Reference signatures were derived prior to deconvolution analysis by computing the average expression of each gene in each cell cluster. To perform deconvolution analysis, ST data, snRNA-seq data and cell-type subpopulation clusters were used as inputs to cell2location. Genes with zero number of counts were removed from the snRNA-seq data and further filtered such that only genes expressed by at least 3% of cells were retained. Furthermore, for genes with high counts in non-zero cells, only those expressed by at least 0.05% of cells were retained. As recommended, this ensured rare cell-type populations were retained but only when expressed at high levels. Cell2location was trained using the following parameters and priors: number of training iterations: 20,000; prior on the expected number of cells per location (N): 8; prior on the expected number of cell-types per location (A): 5; prior on the expected number of co-located cell-type groups per location (Y): 2; prior on the sensitivity of spatial technology: $\mu = 1/2$, $\sigma^2 = 1/8$. Cell2location was also used for spatial co-location analysis of cell subpopulations. This was performed via a non-negative matrix factorisation (NMF) method using the estimated cell abundances as input.

To identify DEGs for each factor, we determined an appropriate minimum expression threshold for spots per factor (herein referred to as hotspots), imported spots above threshold into Loupe browser and computed DEGs versus all spots. Significant spatial genes were subsequently annotated to snRNA-seq expression data, showing the proportion of reads per gene across cell subpopulation clusters as a heatmap.

## Immunohistochemistry

FFPE human biopsy and mouse tissue sections were stained using a standard IHC protocol. Briefly, sections were dewaxed, rehydrated and endogenous peroxidase activity was removed using hydrogen peroxide prior to antigen retrieval by boiling in sodium citrate (pH 6) or Tris-

EDTA (pH 9) depending on manufacturer recommendations for each antibody. Tissue sections were incubated overnight at 4 degrees in a humidified container with the indicated primary antibody; anti-ARHGEF38 (Invitrogen #PA5-57695, 1:300), anti-SOX9 (Merck #ab5535, 1:500), and anti-TROP2 (Abcam #ab214488 1:1000).

After washing, a species-specific biotinylated secondary antibody (VectorLabs) was applied for 2 h at 4 degrees followed by further washes and incubation with Strep-HRP (VectorLabs) for 1 h at 4 degrees. After further washes, colour detection was performed using DAB substrate and toluidine blue was used as a counterstain before dehydration and coverslip mounting in Entellan.

## Imaging mass cytometry (IMC)

Antibodies for imaging mass cytometry were purchased pre-conjugated from Standard BioTools or purchased unconjugated and BSA-free from commercial suppliers and conjugated in-house using the Maxpar® X8 Antibody Labeling Kit (Standard BioTools) following manufacturer recommendations. Platinum (Pt195/196) conjugation used isotopically purified cisplatin Pt195 or Pt196 to label partially reduced antibodies.

FFPE liver tissue sections were sectioned at 5 µm onto Super-frost Plus adhesion slides and dried. Subsequently, slides were dewaxed in xylene and rehydrated through an alcohol gradient to Ultrapure water (Gibco). Slides were then subjected to pH 8.5 Tris-EDTA antigen retrieval (30 min, 96 °C), washed in fresh PBS, blocked in 3% BSA/PBS (45 min, RT) then stained with primary antibody mix (See Appendix Table S1) in 0.5% BSA/PBS (overnight, 4 °C) in a dark humidified chamber. Slides were then washed twice with 0.1% Triton-X/PBS (8 min, RT) and incubated with metal-tagged anti-flurorphore secondary antibodies in 0.5% BSA/PBS (2 h, RT). Slides were washed twice (as before) then incubated with Ir191/193 DNA intercalator (Standard BioTools) 1:400 in PBS (30 min, RT). Slides were then rinsed once with PBS, once with Ultrapure water (Gibco) and finally air-dried.

The region of interest was selected by histopathological features using whole-slide brightfield imaging of H&E-stained sequential sections, defined in CyTOF software (Standard BioTools) and acquired using the Hyperion Imaging Mass Cytometry system (Standard BioTools) calibrated following manufacturer directions. Acquired images were inspected and exported using MCDViewer (Standard BioTools) and analysed using ImageJ.

## In vitro overexpression modelling

HepG2 cells (ATCC HB-8065; routinely screened for mycoplasma) were transfected with either pcDNA3.1 (backbone only control), pcDNA3.1-hSOX9, pFR_HNF6/ONECUT1 (Addgene plasmid #31099), or a combination of SOX9 and HNF6/ ONECUT1 plasmids using a standard Lipofectamine 3000 protocol. 1 µg of total DNA was transfected per well of a 6-well plate. 48 h after transfection, cells were harvested and RNA extracted using an RNEasy extraction kit (Qiagen). 1 µg RNA was converted to cDNA using a high-capacity RNA-to-cDNA kit (Applied Biosystems) prior to quantification of gene expression via RT-qPCR using a SYBR Green based assay on a QuantStudio 1 machine (Applied Biosystems). Relative gene expression was calculated using the delta-delta-Ct method using the average of *β-ACTIN* and *HPRT* housekeeping genes. Primers used were *β-ACTIN* forward

## The paper explained

### Problem

Chronic liver disease (CLD) is a leading cause of morbidity and mortality worldwide, with fibrosis progression culminating in cirrhosis and liver failure. Although potentially reversible, diagnosis is often delayed until advanced stages, and effective antifibrotic treatments remain limited. To address this requires an in depth understanding of the unique microenvironment driving fibrosis that involves complex molecular interactions between multiple cell types.

### Results

In this study we have used a multi-modal approach to uncover gene regulatory networks (GRNs) driving cell state transitions in human liver disease. Data integration from single nuclei snRNA, snATAC and spatial transcriptomic sequencing uncovered the cellular patho-architecture and spatial molecular signatures of diseased liver. Our results highlighted disease associated cell states localised at the scar interface potentially resulting from impaired regeneration in ongoing disease. Through pseudotemporal analysis, we uncovered a directional trajectory and transitional state in these cells between hepatocytes and cholangiocytes. Using our snRNA and ATAC data, we inferred GRNs and identified SOX9/ONECUT1 co-regulation of transitional genes. Through an integrative analysis, we identified a novel transitional axis, marked by ESRP1 (involved in gene splicing) and ARHGEF38 (involved in cell migration) expression, driven by ONECUT1 and SOX9 co-accessibility regulating transitional disease states in cirrhosis.

### Impact

These data serve as a spatial atlas of gene regulation and altered cell states during human liver disease. Our findings highlight novel molecular pathways of fibrosis progression and provide a valuable resource for biomarker discovery and therapeutic targeting in progressive liver disease.

5′-CCAACCGCGAGAAGATGA-3′ and reverse 5′-CCAGAGGCG-TACAGGGATAG-3′, *HPRT* forward 5′-TGACCTTGATTTATTT TGCATACC-3′ and reverse 5′-CGAGCAAGACGTTCAGTCCT-3′, *ARHGEF38* forward 5′-TAGCTGAGACCTTAACCCCAG-3′ and reverse 5′-CACATCCAGCCTATCAGTCTTTT-3′ and *ESRP1* forward 5′-GCCAAGCTAGGCTCGGATG-3′ and reverse 5′-CAGTCCTCCGTCAGTTCCAAC-3′.

## Data availability

The datasets produced in this study are available in ArrayExpress (www.ebi.ac.uk/arrayexpress), with accession numbers E-MTAB-13130 (snRNA-seq), E-MTAB-13131 (snATAC-seq), E-MTAB-13132 (Visium experimental cohort) and E-MTAB-14960 (Visium validation cohort).

The source data of this paper are collected in the following database record: biostudies:S-SCDT-10_1038-S44321-025-00230-6.

## Peer review information

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

## Acknowledgements

This work was supported by the Medical Research Council (MRC; KPH, MR/J003352/1 & MR/P023541/1; NAH, MR/000638/1 & MR/S036121/1). The Genomics Core Facility, Flow Cytometry and the Bioimaging Facility at the University of Manchester are acknowledged for their technical help and support from the Wellcome Trust (105610) and Biotechnology and Biological Sciences Research Council (BBSRC; KNC, BB/S019324/1). This work was supported by the Wellcome Trust Institutional Strategic Support Fund (204796). KPH is a member of the Wellcome Trust supported Centre for Cell-Matrix Research (203128/Z/16/Z) and receives funding from the Centre's Helen Muir Fund.

## Author contributions

**Nigel L Hammond**: Data curation; Formal analysis; Validation; Investigation; Visualization; Methodology; Writing—original draft; Writing—review and editing. **Syed Murtuza Baker**: Data curation; Software; Formal analysis; Investigation; Methodology; Writing—original draft; Writing—review and editing. **Sokratia Georgaka**: Data curation; Software; Formal analysis; Investigation; Methodology; Writing—review and editing. **Ali Al-Anbaki**: Formal analysis; Investigation. **Elliot Jokl**: Formal analysis; Supervision; Investigation; Visualization. **Kara Simpson**: Formal analysis; Investigation. **Rosa Sanchez-Alvarez**: Formal analysis; Investigation. **Varinder S Athwal**: Resources; Formal analysis; Investigation; Writing—original draft; Writing—review and editing. **Huw Purssell**: Resources. **Ajith K Siriwardena**: Resources. **Harry V M Spiers**: Resources. **Mike J Dixo**: Formal analysis; Writing—original draft; Writing—review and editing. **Leoma D Bere**: Formal analysis; Investigation. **Adam P Jones**: Formal analysis; Investigation. **Michael J Haley**: Formal analysis; Investigation. **Kevin N Couper**: Formal analysis; Supervision; Writing—original draft; Writing—review and editing. **Nicoletta Bobola**: Formal analysis; Writing—original draft; Writing—review and editing. **Andrew D Sharrocks**: Formal analysis; Writing—original draft; Writing—review and editing. **Neil A Hanley**: Conceptualization; Formal analysis; Funding acquisition; Writing—original draft; Project administration; Writing—review and editing. **Magnus Rattray**: Conceptualization; Formal analysis; Investigation; Project administration. **Karen Piper Hanley**: Conceptualization; Formal analysis; Supervision; Funding acquisition; Investigation; Visualization; Writing—original draft; Project administration; Writing—review and editing.

Source data underlying figure panels in this paper may have individual authorship assigned. Where available, figure panel/source data authorship is listed in the following database record: biostudies:S-SCDT-10_1038-S44321-025-00230-6.

## Disclosure and competing interests statement

The authors declare no competing interests.

