## [Peer Review File · EMBO Molecular Medicine]

Spatial gene regulatory networks driving cell state transitions during human liver disease

Karen Piper Hanley, Nigel Hammond, Syed Murtuza Baker, Sokratia Georgaka, Ali Al-Anbaki, Elliot Jokl, Kara Simpson, Rosa Sanchez-Alvarez, Varinder Athwal, Huw Purssell, Ajith Siriwardena, Harry Spiers, Michael Dixon, Leoma Bere, Adam Jones, Michael Haley, Kevin Couper, Nicoletta Bobola, Andrew Sharrocks, Neil Hanley, and Magnus Rattray

Corresponding author: Karen Piper Hanley (karen.piperhanley@manchester.ac.uk)

Review Timeline:

Submission Date:	26th Sep 24
Editorial Decision:	24th Oct 24
Revision Received:	16th Feb 25
Editorial Decision:	13th Mar 25
Revision Received:	21st Mar 25
Accepted:	25th Mar 25

Editor: Zeljko Durdevic

Transaction Report:

24th Oct 2024

Dear Prof. Piper Hanley,

Thank you for the submission of your manuscript to EMBO Molecular Medicine. We have now received feedback from the three reviewers who agreed to evaluate your manuscript. All three referees recognize interest of the study but also raise important concerns that should be addressed in a major revision. If you would like to discuss further the points raised by the referees, I am available to do so via email or video. Let me know if you are interested in this option.

We would welcome the submission of a revised version within three months for further consideration. Please let us know if you require longer to complete the revision.

I look forward to receiving your revised manuscript.

Yours sincerely,

Zeljko Durdevic

We require:

- 1) A .docx formatted version of the manuscript text (including legends for main figures, EV figures and tables). Please make sure that the changes are highlighted to be clearly visible.
- 2) Individual production quality figure files as .eps, .tif, .jpg (one file per figure). For guidance, download the 'Figure Guide PDF': (<https://www.embopress.org/page/journal/17574684/authorguide#figureformat>).
- 3) A .docx formatted letter INCLUDING the reviewers' reports and your detailed point-by-point responses to their comments. As part of the EMBO Press transparent editorial process, the point-by-point response is part of the Review Process File (RPF), which will be published alongside your paper.
- 4) A complete author checklist, which you can download from our author guidelines (<https://www.embopress.org/page/journal/17574684/authorguide#submissionofrevisions>). Please insert information in the checklist that is also reflected in the manuscript. The completed author checklist will also be part of the RPF.
- 5) Please note that all corresponding authors are required to supply an ORCID ID for their name upon submission of a revised manuscript.
- 6) It is mandatory to include a 'Data Availability' section after the Materials and Methods. Before submitting your revision, primary datasets produced in this study need to be deposited in an appropriate public database, and the accession numbers and

database listed under 'Data Availability'. Please remember to provide a reviewer password if the datasets are not yet public (see <https://www.embopress.org/page/journal/17574684/authorguide#dataavailability>).

12) Author contributions: You will be asked to provide CRediT (Contributor Role Taxonomy) terms in the submission system. These replace a narrative author contribution section in the manuscript.

13) A Conflict of Interest statement should be provided in the main text.

14) Every published paper now includes a 'Synopsis' to further enhance discoverability. Synopses are displayed on the journal webpage and are freely accessible to all readers. They include a short stand first (maximum of 300 characters, including space) as well as 2-5 one-sentences bullet points that summarizes the paper. Please write the bullet points to summarize the key NEW findings. They should be designed to be complementary to the abstract - i.e. not repeat the same text. We encourage inclusion of key acronyms and quantitative information (maximum of 30 words / bullet point). Please use the passive voice. Please attach

these in a separate file or send them by email, we will incorporate them accordingly.

15) Include a Reagents and Tools Table as part of the Methods section, which can be downloaded from our author guidelines (<https://www.embopress.org/page/journal/17574684/authorguide#structuredmethods>)

***** Reviewer's comments *****

Referee #1 (Comments on Novelty/Model System for Author):

The data could be very informative and valuable for many researchers in the area. The work is all on human tissue which makes it relevant for medical impact.

Referee #1 (Remarks for Author):

Summary:

This manuscript describes the spatial, transcriptomic and epigenetic differences between healthy and fibrotic/cirrhotic liver tissue. The primary goal is to uncover molecular pathways underlying fibrosis in order to improve detection and inform the design of effective therapies. Spatial single-cell transcriptomics, coupled with single-nuclei RNA-seq and ATAC-seq datasets from human healthy and cirrhotic livers were utilized. Analysis of these datasets identified an exhaustive catalogue of cell types in these tissues, as well as their associated expression markers. The authors identify that certain cell subtypes are preferentially organized at the fibrotic tissue interface creating a fibrotic niche, which include parenchymal and non-parenchymal cells. Their analysis also identify a hepatocyte population, along with associated genes and regulatory sequences, that is in a potentially transitional state to a biliary fate as a result of injury and fibrosis.

Impact:

Overall, this study provides invaluable datasets to the scientific community focused on understanding the differences in the gene expression dynamics and regulatory chromatin regions that accompany scar formation. The authors provide a meticulous description of the cell types identified in their study and their spatial distribution in liver tissue in both healthy and disease states. The datasets will be an important resource for the community.

This study however, requires some additional support to bolster some of the novel claims made throughout the manuscript (as described in the comments and suggestions below). Most of the manuscript is descriptive of the dataset, and perhaps could spend more effort investigating the mechanisms underlying the differences between healthy and fibrotic tissue/cell populations. This also includes conducting supportive experiments. It is also unclear why the two genes which are explored in detail, ERP1 and ARHGEF38 are chosen for follow up.

Comments and Suggestions:

1. The citations to Supplementary Figures need to be fixed. Several point to Supplementary Figures that do not appear to be associated with what is written in the text. An example is on line 723 which summarizes validation cohort sample sequencing metrics and cites Supplementary Figure 7, however that figure does not contain ST data for the validation cohort. This occurs frequently throughout the manuscript making it confusing to read.
2. In the initial ST analysis of scar tissue in 'sample b', the expression in cluster b5 (a scar interface cluster) show an noticeable increase in expression of interface gene markers (including cholangiocyte markers) compared to b2. The manuscript would benefit from some commentary from the authors regarding the reason for this difference.
3. In Supplementary Figures 3 (initial cohort) and 4 (validation cohort), ST analysis indicate that specific clusters for the parenchyma, scar and/or discrete scar regions are detected in each sample. Are the genes highly expressed in these clusters correlated between patient samples? Or are they similar to clusters identified in 'sample b'? An analysis of the similarities and differences between clusters located in similar tissue between would help bolster the idea of common gene expression programs in fibrotic tissue across different patients, as it would appear they have different pathology.
4. The authors show a correlation between promoter and gene expression for specific markers to conclude that their ATAC and RNA-seq modalities are similar, however do not show the correlation across genes and promoters. It would be interesting to see how the correlation applies to all expressed genes.
5. Line 187-189: This conclusion requires clarification and support. Are the authors claiming that there are a larger proportion of

LSECs (EC1-3) derived from healthy tissue in their analysis compared to cirrhotic tissue? Additionally, is the conclusion that EC4-6, containing vascular ECs, more abundant in cirrhotic tissue. If so, the proportion of each cell type and the samples from which they were derived would be required to support this claim. And if it is true, could the authors comment on the potential reasons?

6. Since there is a difference in localization in EC and MCs between healthy and cirrhotic liver tissues, it would be interesting to investigate if there are any differences in gene expression or chromatin accessibility in those cell type clusters the two phenotypes.

7. To help support the conclusions of differences in localization in fibrotic livers of certain cell types, quantitative representation of the gene expression or number cells per cluster in either a scar or normal tissue would be beneficial. A quantitative representation (bar plot or box plots) would support the qualitative assessment displayed in spatial deconvolution maps throughout the paper.

8. The authors conclude that HC1 consists of hepatocytes that may be expressing a disease-associated profile. To support this claim, the proportion of cells from healthy and cirrhotic tissue in HC1 could be shown.

9. Some factors expressed in the fibrotic niche (FN1 and VIM) are SOX9 regulated. suggesting that SOX9 is important for gene regulation in these regions. It would be interesting to explore the snATAC-seq data for cells predicted to be present in this fibrotic niche and explore whether SOX9 is enriched at putative regulatory regions, or regions that are specifically accessible in these cells compared to those found in healthy tissue.

10. The authors claim that their RNA velocity indicate that there is a trans-differentiation occurring in HC1 hepatocytes to a more cholangiocyte profile due the shift in expression. However, to support this claim, biological validation is required. An example could be to prioritize transitional transcription factors that are upregulated in HC1/CC1 populations (ONECUT1) and overexpress these factors in normal hepatocytes and measure the gene expression of factors that may change in a disease state or are expressed in a biliary cell fate

11. The authors state, "we have provided a publicly available map and resource for further integrative studies requiring insight from single cell transcriptomics, epigenomics and spatial gene expression in human liver." However, I was unable to find a description of where the information is available. To aid accessibility, a web portal could be created to query genes or chromatin regions of interest and explore their expression in the spatial or single cell space.

12. Paragraph breaks at line 198 and 215 would be beneficial.

13. "Similar to our previous work on localisation of SOX9 and 337 EPCAM in human liver development and disease-associated hepatocytes"

Referee #2 (Remarks for Author):

Nigel L Hammond and co-authors provide massive data analysis on fibrotic and cirrhotic liver tissue. The authors have used human and mouse tissue, several state of the art techniques in spatial transcriptomics and RNAseq and cell culture work. The results provide a lot of information on various cell types and their cellular neighbourhood. However, to my feeling, most data reported are not new but more provide a summary and the confirmation of things already known. Identified markers such as sma or vimentin have been investigated extensively so I miss a bit the novelty of the manuscript. Maybe the authors can work on that point to better highlight their findings and the clinical impact.

Furthermore I have several comments:

- Figure 1: is this fibrosis or cirrhosis? To my feeling it is cirrhosis not fibrosis as indicated. Maybe this should be reconfirmed with a pathologist as highly relevant for discussion and result interpretation.

- Figure 2: what does "mesenchymal cells" mean? Fibroblasts, hepatic stellate cells, pericytes, endothelial cells? Please clarify.

- Figure 2: B cells are lymphocytes, why is there a separate listing of B cell and not T cells?

- Human samples: more clinical data are needed: what is the underlying disease of fibrosis/cirrhosis? The results need to be discussed in context of the underlying disease. Furthermore, it sounds like that samples from tumor patients have been used. If so the authors need to include data on non-tumorous patients as cancer infiltration (even far away from the tumor) can influence the microenvironment.

- Visium: how sure can the authors be that with Visium/space Ranger a single cell resolution can be achieved? In our hands, this doesn't work very well. maybe a section on this in the discussion part could be added.

Referee #3 (Comments on Novelty/Model System for Author):

In this paper Hammond et al present data from snRNAseq, spatial transcriptome and scATAC-seq experiments in human liver samples from patients with cirrhosis. The study identified spatial gene signatures of fibrotic scars and damaged parenchymal cells and a defect in the regenerative response at the hepatocyte-fibrotic scar interface.

The data as presented are highly descriptive and therefore the paper is considered as a "resource study", which provides important information for future functional analyses. It is a very thorough and high-quality work. It combines low resolution spatial transcriptome assessment with high resolution snRNAseq. The analysis tools used are well established and reliable. Apart from the wealth of useful information provided about the spatial distribution and heterogeneity of the different cell types in the liver, the authors also demonstrate the existence of disease-specific variant hepatocyte and cholangiocyte subtypes. Using RNA velocity analyses the authors identified transition states, which could be relevant to the mechanism of fibrosis progression. In addition, the authors provide analyses that revealed dynamic changes in key transcription factors in the disease specific hepatocyte cell states at the scar interface.

Overall, this is a nicely organized and important resource study.

Minor questions:

- 1) The trajectory analysis and integrative pseudotime on snATAC-seq in Figure 8, nicely demonstrates the transition of HC1 to CC1. The finding is consistent with previous studies proposing the transdifferentiation potential of hepatocytes to cholangiocytes during regeneration. The question arises whether the trajectory can be demonstrated also in the opposite direction.
- 2) The hepatic TF dynamics in the different hepatocyte cell states at the interface with the fibrotic regions is remarkable. Although one can see the differences in the heatmap of Figure 9, a line graph version for the major TFs (HNF4, NR2F6, NR4A2, ONECUT1) is advised to better highlight the differences. The feature plot at panel B in Fig 9 for HNF4 and ONECUT1 is puzzling. It seems that substantial HNF4 signal exists in CC cell types and ONECUT signal is stronger in HC cell types.
- 3) In the browser pictures of Fig. 9D the co-accessible enhancer-promoter interactions are probably called by Peak-to-gene link analysis. It would be good to have a global view, e.g. comparisons in Venn diagrams about the existence of co-accessible regions identified in HCs in CCs and vice versa.

Referee #1 (Comments on Novelty/Model System for Author):

The data could be very informative and valuable for many researchers in the area. The work is all on human tissue which makes it relevant for medical impact.

Referee #1 (Remarks for Author):

Summary:

This manuscript describes the spatial, transcriptomic and epigenetic differences between healthy and fibrotic/cirrhotic liver tissue. The primary goal is to uncover molecular pathways underlying fibrosis in order to improve detection and inform the design of effective therapies. Spatial single-cell transcriptomics, coupled with single-nuclei RNA-seq and ATAC-seq datasets from human healthy and cirrhotic livers were utilized. Analysis of these datasets identified an exhaustive catalogue of cell types in these tissues, as well as their associated expression markers. The authors identify that certain cell subtypes are preferentially organized at the fibrotic tissue interface creating a fibrotic niche, which include parenchymal and non-parenchymal cells. Their analysis also identify a hepatocyte population, along with associated genes and regulatory sequences, that is in a potentially transitional state to a biliary fate as a result of injury and fibrosis.

Impact:

Overall, this study provides invaluable datasets to the scientific community focused on understanding the differences in the gene expression dynamics and regulatory chromatin regions that accompany scar formation. The authors provide a meticulous description of the cell types identified in their study and their spatial distribution in liver tissue in both healthy and disease states. The datasets will be an important resource for the community.

We thank the reviewer for their comments.

This study however, requires some additional support to bolster some of the novel claims made throughout the manuscript (as described in the comments and suggestions below). Most of the manuscript is descriptive of the dataset, and perhaps could spend more effort investigating the mechanisms underlying the differences between healthy and fibrotic tissue/cell populations. This also includes conducting supportive experiments. It is also unclear why the two genes which are explored in detail, ERP1 and ARHGEF38 are chosen for follow up.

Apologies, ESRP1 and ARHGEF38 were amongst our highest differentially expressed genes as part of the HC1 cluster. We have added a sentence to explain this.

Comments and Suggestions:

1. The citations to Supplementary Figures need to be fixed. Several point to Supplementary Figures that do not appear to be associated with what is written in the text. An example is on line 723 which summarizes validation cohort sample sequencing metrics and cites Supplementary Figure 7, however that figure does not contain ST data for the validation cohort. This occurs frequently throughout the manuscript making it confusing to read.

Apologies and thank you. We have fixed this problem which related to the order of the figures in the Supplementary Information rather than their citation in the manuscript.

2. In the initial ST analysis of scar tissue in 'sample b', the expression in cluster b5 (a scar interface cluster) show an noticeable increase in expression of interface gene markers (including cholangiocyte markers) compared to b2. The manuscript would benefit from some commentary from the authors regarding the reason for this difference.

The markers identified in cluster b5 are typical of cholangiocyte markers, and while generally the expression of these genes are higher within the scar (b2) there are 'hotspots' which show significant enrichment (b5). These discrete regions highlighted by b5 likely identify reactive bile ducts. The local enrichment of these markers are highlighted further as cluster b5 consists of 28 spots versus 222 spots for cluster b2. We have made some commentary regarding this in the results.

3. In Supplementary Figures 3 (initial cohort) and 4 (validation cohort), ST analysis indicate that specific clusters for the parenchyma, scar and/or discrete scar regions are detected in each sample. Are the genes highly expressed in these clusters correlated between patient samples? Or are they similar to clusters identified in 'sample b'? An analysis of the similarities and differences between clusters located in similar tissue between would help bolster the idea of common gene expression programs in fibrotic tissue across different patients, as it would appear they have different pathology.

The differing pathology in our fibrotic patient biopsy tissue reflects the severity of the disease. Samples D & E are moderate fibrosis whereas F & G are severe fibrosis (see dataset EV1 for patient details and fibrotic pathology staging). Although direct comparisons between different spatial technologies is challenging with more divergent spatial gene signatures (i.e. frozen used for the initial cohort v FFPE in the validation cohort) there are similar themes. For example, in exploring this point we compared the top genes expressed in 'scar-associated' Louvain clusters per sample, across all patients A-G (Dataset EV7; with a notes section explaining how the data was generated). Common signatures were most prevalent in the MC populations with a noticeable difference in signatures associated with inflammatory cells. We have added this to our results.

4. The authors show a correlation between promoter and gene expression for specific markers to conclude that their ATAC and RNA-seq modalities are similar, however do not show the correlation across genes and promoters. It would be interesting to see how the correlation applies to all expressed genes.

Thank you for this helpful comment. We have provided further insight by correlating peaks to genes using both modalities (Appendix Figure S6A). We also show the proportion of HC and CC peaks which are predicted to regulate the same genes (addressing Reviewer 3; Appendix Figure S6B).

5. Line 187-189: This conclusion requires clarification and support. Are the authors claiming that there are a larger proportion of LSECs (EC1-3) derived from healthy tissue in their analysis compared to cirrhotic tissue? Additionally, is the conclusion that EC4-6, containing vascular ECs, more abundant in cirrhotic tissue. If so, the proportion of each cell type and the samples from which they were derived would be required to support this claim. And if it is true, could the authors comment on the potential reasons?

Apologies if this was not clear, we are primarily making an observation rather than a claim regarding cell type abundance in healthy vs disease. Our data shows that healthy and cirrhotic LSECs overlap, whereas EC4-6 are more distinct from healthy cells in the UMAP. This is potentially due to elevated disease-specific signatures within EC sub-clusters (particularly EC4, EC5) (see dot plot Appendix Figure S9C). We have toned down the language in the results.

6. Since there is a difference in localization in EC and MCs between healthy and cirrhotic liver tissues, it would be interesting to investigate if there are any differences in gene expression or chromatin accessibility in those cell type clusters the two phenotypes.

Apologies if we have misinterpreted this point; we think this relates to our spatial mapping of the 'cirrhotic EC and MC signatures' (Appendix Figures S9E,F and S11E,F). The localisation of EC/MC are the same. However, within the scar, sub-populations of EC and MC appear to have different gene

expression profiles most likely because of the profibrotic microenvironment. It is these disease-specific signatures (healthy vs cirrhotic DEGs) we are plotting.

7. To help support the conclusions of differences in localization in fibrotic livers of certain cell types, quantitative representation of the gene expression or number cells per cluster in either a scar or normal tissue would be beneficial. A quantitative representation (bar plot or box plots) would support the qualitative assessment displayed in spatial deconvolution maps throughout the paper.

Thank you for this suggestion. We have quantified the fibrotic cell-type signatures within Visium validation samples by quantifying log normalised UMIs in scar-associated and non-scar regions, displayed as violin plots (Appendix Figure S10).

8. The authors conclude that HC1 consists of hepatocytes that may be expressing a disease-associated profile. To support this claim, the proportion of cells from healthy and cirrhotic tissue in HC1 could be shown.

HC1 is a unique cirrhotic population. However, to address this point we have re-clustered the integrated healthy and disease datasets (Leiden) to show the cell type contribution within parenchymal cells (Appendix Figure S14C & D). This shows HC1 is unique (99% contribution of cirrhotic cells to cluster 8), with 3-4% contribution at most for HC1 cells to other HC clusters. The dot plots for these data (Appendix Figure S14A), further confirm this for gene expression. Additionally, we have provided more detailed information on the sample and cell type contribution to the integrated healthy and cirrhotic snRNA-seq data (Appendix Figure S7). In combination, we hope this helps to support our work.

9. Some factors expressed in the fibrotic niche (FN1 and VIM) are SOX9 regulated, suggesting that SOX9 is important for gene regulation in these regions. It would be interesting to explore the snATAC-seq data for cells predicted to be present in this fibrotic niche and explore whether SOX9 is enriched at putative regulatory regions, or regions that are specifically accessible in these cells compared to those found in healthy tissue.

Although this would be interesting to explore, the publically available healthy data sets used in this study do not have paired ATACseq data – limiting our ability to fully address this in the current manuscript. We feel this would be better addressed with a new study as our own data and published data from healthy liver become available.

10. The authors claim that their RNA velocity indicate that there is a trans-differentiation occurring in HC1 hepatocytes to a more cholangiocyte profile due the shift in expression. However, to support this claim, biological validation is required. An example could be to prioritize transitional transcription factors that are upregulated in HC1/CC1 populations (ONECUT1) and overexpress these factors in normal hepatocytes and measure the gene expression of factors that may change in a disease state or are expressed in a biliary cell fate

To support our biology we have already provided in vivo expression and in vitro over-expression of SOX9 and OC1 in HepG2 cells to gain insight on our specific factors ESRP1 and ARHGEF38 (Appendix Figure S24). Apologies if this was overlooked. Our insight indicates that transient overexpression of a single TF is inadequate to force a full phenotypic cell type change (i.e. induce a cholangiocyte from a hepatocyte) particularly in vitro. However, we are not expecting a full cell type shift as this disease cell state phenotype appears to be somewhere in between.

11. The authors state, "we have provided a publicly available map and resource for further integrative studies requiring insight from single cell transcriptomics, epigenomics and spatial gene expression in human liver." However, I was unable to find a description of where the information is

available. To aid accessibility, a web portal could be created to query genes or chromatin regions of interest and explore their expression in the spatial or single cell space.

We have provided a link to our online resource in the introduction and discussion (https://cellxgene.bmh.manchester.ac.uk/liver_disease/)

12. Paragraph breaks at line 198 and 215 would be beneficial.

Thank you – we have inserted these breaks.

13. "Similar to our previous work on localisation of SOX9 and 337 EPCAM in human liver development and disease-associated hepatocytes"

Thank you – we have amended this.

Referee #2 (Remarks for Author):

Nigel L Hammond and co-authors provide massive data analysis on fibrotic and cirrhotic liver tissue. The authors have used human and mouse tissue, several state of the art techniques in spatial transcriptomics and RNAseq and cell culture work. The results provide a lot of information on various cell types and their cellular neighbourhood. However, to my feeling, most data reported are not new but more provide a summary and the confirmation of things already known. Identified markers such as sma or vimentin have been investigated extensively so I miss a bit the novelty of the manuscript. Maybe the authors can work on that point to better highlight their findings and the clinical impact.

We thank the reviewer for these comments and sorry they have missed the novelty particularly given the comments from the other two reviewers on the impact and importance of the data.

Furthermore I have several comments:

- Figure 1: is this fibrosis or cirrhosis? To my feeling it is cirrhosis not fibrosis as indicated. Maybe this should be reconfirmed with a pathologist as highly relevant for discussion and result interpretation.

Please see the methods describing the human tissue. All of the samples in Fig 1 are clinically cirrhotic as indicated and confirmed by a histopathologist as described. In the results the tissue is discussed as being from cirrhotic individuals. From discussion with the hepatology team on the manuscript (and colleagues outside of Manchester) fibrosis appears to be valid terminology while discussing the pathological scar in a clinically cirrhotic patient sample. The data we have gained from the experimental cohort was used to explore signatures spatially in further patients with fibrosis ranging from IS3 (moderate fibrosis) through to IS6 (severe fibrosis/cirrhosis). We have revised wording in the results, discussion and Figure 1 legend title to help clarify.

- Figure 2: what does "mesenchymal cells" mean? Fibroblasts, hepatic stellate cells, pericytes, endothelial cells? Please clarify.

We have used broad terminology as the cellular marker base aligns with mesenchymal origins. We provide a full explanation of each subcluster to make this clear in the results. Mesenchymal cells refers to hepatic stellate cells (MC1) and vascular smooth muscle cells (MC2, potentially pericytes). Endothelial cells (ECs) are specifically labelled and are not mesenchymal cells. This is all detailed in the results.

- Figure 2: B cells are lymphocytes, why is there a separate listing of B cell and not T cells?

Apologies for the confusion, for continuity we have renamed this population LC3.

- Human samples: more clinical data are needed: what is the underlying disease of fibrosis/cirrhosis? The results need to be discussed in context of the underlying disease. Furthermore, it sounds like that samples from tumor patients have been used. If so the authors need to include data on non-tumorous patients as cancer infiltration (even far away from the tumor) can influence the microenvironment.

The clinical data is provided in Dataset EV1 on all of the validation cohort biopsy samples used in this project. The methodology section provides all clinical information on the experimental cohort, we have now added the background diagnosis which was MASLD as requested.

Both the experimental discovery and validation cohort have a background of MASLD. We have added insight into the clinical impact in the discussion where these datasets may provide insight into genetic variants and causal genes associated with the disease.

We are aware of the potential influence of the HCC environment and for this specific reason we included a validation cohort, with spatial data and to validate our disease associated signatures from our experimental discovery cohort. These biopsy samples are from non-cancerous patients (full clinical data can be found in Dataset EV1) with fibrosis ranging from IS3 – IS6.

- Visium: how sure can the authors be that with Visium/space Ranger a single cell resolution can be achieved? In our hands, this doesn't work very well. maybe a section on this in the discussion part could added.

We do not state that spatial transcriptomics gives single cell resolution and further mention/reinforce this at the start of the results section ' However, information on the cell-type contribution to these signatures was obfuscated by the current resolution of the spatial spots'. Full methods support this and describe the platform used.

Referee #3 (Comments on Novelty/Model System for Author):

In this paper Hammond et al present data from snRNAseq, spatial transcriptome and scATAC-seq experiments in human liver samples from patients with cirrhosis. The study identified spatial gene signatures of fibrotic scars and damaged parenchymal cells and a defect in the regenerative response at the hepatocyte-fibrotic scar interface.

The data as presented are highly descriptive and therefore the paper is considered as a "resource study", which provides important information for future functional analyses. It is a very thorough and high-quality work. It combines low resolution spatial transcriptome assessment with high resolution snRNAseq. The analysis tools used are well established and reliable. Apart from the wealth of useful information provided about the spatial distribution and heterogeneity of the different cell types in the liver, the authors also demonstrate the existence of disease-specific variant hepatocyte and cholangiocyte subtypes. Using RNA velocity analyses the authors identified transition states, which could be relevant to the mechanism of fibrosis progression. In addition, the authors provide analyses that revealed dynamic changes in key transcription factors in the disease specific hepatocyte cell states at the scar interface.

Overall, this is a nicely organized and important resource study.

We thank the reviewer for their comments.

Minor questions:

1) The trajectory analysis and integrative pseudotime on snATAC-seq in Figure 8, nicely demonstrates the transition of HC1 to CC1. The finding is consistent with previous studies proposing

the transdifferentiation potential of hepatocytes to cholangiocytes during regeneration. The question arises whether the trajectory can be demonstrated also in the opposite direction.

Thank you for this comment. We have been careful to not overstate the directionality as this could indeed be bi-directional. The analysis we have used would be the same in the opposite direction. We make assumptions on the direction, but the underlying data does not change whichever direction is chosen.

2) The hepatic TF dynamics in the different hepatocyte cell states at the interface with the fibrotic regions is remarkable. Although one can see the differences in the heatmap of Figure 9, a line graph version for the major TFs (HNF4, NR2F6, NR4A2, ONECUT1) is advised to better highlight the differences. The feature plot at panel B in Fig 9 for HNF4 and ONECUT1 is puzzling. It seems that substantial HNF4 signal exists in CC cell types and ONECUT signal is stronger in HC cell types.

Apologies and thank you for highlighting this mistake. The plots in 9B were generated by Cell Ranger algorithms which do not fully represent the integrated modality ArchR data displayed for Figure 9A. We apologise for this discrepancy and have replaced these plots with motif deviation UMAPs and line plots, generated with ArchR, for the main TFs of interest.

3) In the browser pictures of Fig. 9D the co-accessible enhancer-promoter interactions are probably called by Peak-to-gene link analysis. It would be good to have a global view, e.g. comparisons in Venn diagrams about the existence of co-accessible regions identified in HCs in CCs and vice versa.

To address this and comments from Reviewer 1, we have provided peak to gene correlations for snATAC-seq/snRNA-seq genome-wide across all cell types (Appendix Figure S6A). In addition, we show that 27% of HC/CC enhancers are co-accessible in both cell types (Appendix Figure S6B).

13th Mar 2025

Dear Prof. Piper Hanley,

Thank you for the submission of your revised manuscript to EMBO Molecular Medicine. As you will see from their reports pasted below, while referees #1 and #3 support publication of the manuscript, referee #2 recognizes interest of the study but also raises important concerns. Based on the initial referee reports and after an editorial discussion, we agreed that the authors responded adequately to the referees' criticism. Therefore, I am pleased to inform you that we will be able to accept your manuscript pending the following final amendments:

- 1) Please address all referee #2 concerns. No additional experiments are required. Please check the manuscript and make sure that all information are correct in the results, methods and figure legends.
- 2) In the main manuscript file, please do the following:
 - Please address all comments suggested by our data editors listed below:
 - o Data availability statement:
 1. Please note that the specific URLs for E-MTAB-13130, EMTAB-13131, E-MTAB-13132 datasets are not provided in the data availability statement.
 - Add up to 5 keywords.
 - Add callouts for the Figure 2A.
 - Provide the antibody dilutions that were used for each antibody.
 - In Methods, provide the statement that informed consent was obtained from all human subjects and confirm that the experiments conformed to the principles set out in the WMA Declaration of Helsinki and the Department of Health and Human Services Belmont Report.
 - Author contributions: Please remove it from the manuscript and specify author contributions in our submission system. CRediT has replaced the traditional author contributions section because it offers a systematic machine-readable author contributions format that allows for more effective research assessment. You are encouraged to use the free text boxes beneath each contributing author's name to add specific details on the author's contribution. More information is available in our guide to authors:
<https://www.embopress.org/page/journal/17574684/authorguide#authorshipguidelines>
 - Indicate in legends exact n and exact p values, not a range, along with the statistical test used. To keep the figures "clear" some authors found providing an Appendix table Sx with all exact p-values preferable. You are welcome to do this if you want to.
- 3) Datasets: Please submit Dataset EV1 as Table EV1. Rest of the files should be renamed to Dataset EV1-EV8. Also, update their callouts in the main text.
- 4) Appendix: Please move "Appendix Information" to the Results section and leave only figures and tables in the Appendix. Add page numbers to the table of content and upload the file in PDF format.
- 5) Funding: Please make sure that information about all sources of funding are complete in both our submission system and in the manuscript. Currently, Wellcome Trust Institutional Strategic Support Fund (204796) and the Helen Muir Fund are missing in our system.
- 6) The Paper Explained: Please add it to the main manuscript file.
- 7) Synopsis: Please check your synopsis text and image before submission with your revised manuscript. Please be aware that in the proof stage minor corrections only are allowed (e.g., typos).
- 8) Source data: Please upload original, unprocessed, individual microscopic images zipped in one folder for Figure 7.
- 9) As part of the EMBO Publications transparent editorial process initiative (see our Editorial at <http://embomolmed.embopress.org/content/2/9/329>), EMBO Molecular Medicine will publish online a Review Process File (RPF) to accompany accepted manuscripts. This file will be published in conjunction with your paper and will include the anonymous referee reports, your point-by-point response and all pertinent correspondence relating to the manuscript. Let us know whether you agree with the publication of the RPF and as here, if you want to remove or not any figures from it prior to publication. Please note that the Authors checklist will be published at the end of the RPF.
- 10) Please provide a point-by-point letter INCLUDING my comments as well as the reviewer's reports and your detailed responses (as Word file).

I look forward to reading a new revised version of your manuscript as soon as possible.

Yours sincerely,

Zeljko Durdevic

*** Instructions to submit your revised manuscript ***

- 1) a .docx formatted version of the manuscript text (including Figure legends and tables)
 - 2) Separate figure files*
 - 3) supplemental information as Expanded View and/or Appendix. Please carefully check the authors guidelines for formatting Expanded view and Appendix figures and tables at <https://www.embopress.org/page/journal/17574684/authorguide#expandedview>
 - 4) a letter INCLUDING the reviewer's reports and your detailed responses to their comments (as Word file).
 - 5) The paper explained: EMBO Molecular Medicine articles are accompanied by a summary of the articles to emphasize the major findings in the paper and their medical implications for the non-specialist reader. Please provide a draft summary of your article highlighting
 - the medical issue you are addressing,
 - the results obtained and
 - their clinical impact.This may be edited to ensure that readers understand the significance and context of the research. Please refer to any of our published articles for an example.
 - 6) Author contributions: the contribution of every author must be detailed in a separate section.
 - 7) EMBO Molecular Medicine now requires a complete author checklist (<https://www.embopress.org/page/journal/17574684/authorguide>) to be submitted with all revised manuscripts. Please use the checklist as guideline for the sort of information we need WITHIN the manuscript. The checklist should only be filled with page numbers where the information can be found. This is particularly important for animal reporting, antibody dilutions (missing) and exact values and n that should be indicated instead of a range.
 - 8) Every published paper now includes a 'Synopsis' to further enhance discoverability. Synopses are displayed on the journal webpage and are freely accessible to all readers. They include a short stand first (maximum of 300 characters, including space) as well as 2-5 one sentence bullet points that summarise the paper. Please write the bullet points to summarise the key NEW findings. They should be designed to be complementary to the abstract - i.e. not repeat the same text. We encourage inclusion of key acronyms and quantitative information (maximum of 30 words / bullet point). Please use the passive voice. Please attach these in a separate file or send them by email, we will incorporate them accordingly.
- You are also welcome to suggest a striking image or visual abstract to illustrate your article. If you do please provide a jpeg file 550 px-wide x 300-600px high.
- 9) A Conflict of Interest statement should be provided in the main text
 - 10) Please note that we now mandate that all corresponding authors list an ORCID digital identifier. This takes <90 seconds to complete. We encourage all authors to supply an ORCID identifier, which will be linked to their name for unambiguous name identification.

Currently, our records indicate that the ORCID for your account is 0000-0001-9473-9647.

Please click the link below to modify this ORCID:
Link Not Available

11) Include a Reagents and Tools Table as part of the Methods section, which can be downloaded from our author guidelines (<https://www.embopress.org/page/journal/17574684/authorguide#structuredmethods>)

Graphs 800-1,200 DPI
Photos 400-800 DPI
Colour (only CMYK) 300-400 DPI"

*Additional important information regarding figures and illustrations can be found at <https://bit.ly/EMBOPressFigurePreparationGuideline>. See also figure legend preparation guidelines: <https://www.embopress.org/page/journal/17574684/authorguide#figureformat>

**** Reviewer's comments ****

Referee #1 (Comments on Novelty/Model System for Author):

The data could be very informative and valuable for many researchers in the area. The work is all on human tissue which makes it relevant for medical impact.

Referee #1 (Remarks for Author):

The authors have sufficiently addressed our comments. We believe this manuscript has a lot of interesting information.

Referee #2 (Remarks for Author):

The technical quality of the manuscript is very high. The authors present truly massive data from various spatial techniques and detailed results on liver fibrosis and I have absolutely no doubt about the technical quality of the manuscript. The novelty to me is only medium as most of the factors and molecules identified are well known however the spatial resolution and composition is new and very interesting. Still, I have some concerns about the results and the interpretation based on the samples the authors used. I don't think that a surgical pathologist or someone with advanced knowledge in histology has been involved in the manuscript which and to my feeling this would be absolutely necessary to link spatial genetic data to morphology: E.g. it was shown in EV1 (excel file) that most samples are derived from patients with history of alcohol or MASLD. Both diagnosis typically show fatty liver disease (as stated by the S in MASLD) however NONE of the samples provided in the whole manuscript shows steatosis in the given H&E stainings. This raises the question if the pictures were not representative, the etiology was wrong or the samples just didn't contain fat which to my understanding then changes the interpretation a lot. Further, in the figure legend of Appendix Figure S1 it says that three samples of cirrhotic livers are shown. However, sample c (and again k in Appendix figure S3) is not cirrhotic, does not even display advanced fibrosis but just portal fibrosis. Interestingly, in the main text of the manuscript it was described correctly (see line 93-95). Was this just a mistake or did the authors work with sample C as cirrhotic liver? Again, given the low number of samples this can change the interpretation a lot. Further, I don't think that the HIF1alpha staining in figure 7 worked very well. Normally HIF 1alpha isn't that patchy but more diffuse. Did the authors include proper controls or how can they assume that this staining really worked?

Further for large parts of the manuscript, a validation on the protein level is unfortunately missing. The authors present some data e.g. on SOX9 (immunohistochemistry) which is great but for several other exciting findings (such as atypical HC1 showing overlapping gene signatures with CC1 in Fig 5 D, E or Appendix Figure S10 A) it would be absolutely necessary to have a validation on the protein level e.g. by immunohistochemistry or immunofluorescence to understand the biological significance.

Same for Figure 1 G,K - here, it would be very nice to highlight the findings in an additional cytokeratin 7 immunostaining to show the bile ducts and their morphology as bile duct hyperplasia is a common finding in cirrhotic liver and nothing new.

I think all this issues need to clarified as this work might probably be considered as a resource study with a high likelihood to get cited.

Therefore don't think the manuscript given in its present form is appropriate for publication.

Referee #3 (Remarks for Author):

I am satisfied with the answers to my concerns. I believe that the results, especially those describing the transition states and the molecular signatures associated with them in the fibrotic niche, will have a significant impact for the field.

We thank the editor for their comments and handling our manuscript. We have addressed all points above as requested. We have highlighted any new changes in the submitted manuscript. We have ensured the wording regarding the experimental cohort is described as 'tissue from patients diagnosed with cirrhosis' rather than 'cirrhotic tissue' where appropriate in results, methods and figure legends (including supplementary legends). We have taken the editors comments into context while addressing reviewer 2's comments below.

***** Reviewer's comments *****

Referee #1 (Comments on Novelty/Model System for Author):

The data could be very informative and valuable for many researchers in the area. The work is all on human tissue which makes it relevant for medical impact.

Referee #1 (Remarks for Author):

The authors have sufficiently addressed our comments. We believe this manuscript has a lot of interesting information.

Thank you for your time assessing our manuscript. We are very grateful.

Referee #2 (Remarks for Author):

The technical quality of the manuscript is very high. The authors present truly massive data from various spatial techniques and detailed results on liver fibrosis and I have absolutely no doubt about the technical quality of the manuscript. The novelty to me is only medium as

most of the factors and molecules identified are well known however the spatial resolution and composition is new and very interesting.

Still, I have some concerns about the results and the interpretation based on the samples the authors used. I don't think that a surgical pathologist or someone with advanced knowledge in histology has been involved in the manuscript which and to my feeling this would be absolutely necessary to link spatial genetic data to morphology: E.g. it was shown in EV1 (excel file) that most samples are derived from patients with history of alcohol or MASLD. Both diagnosis typically show fatty liver disease (as stated by the S in MASLD) however NONE of the samples provided in the whole manuscript shows steatosis in the given H&E stainings. This raises the question if the pictures were not representative, the etiology was wrong or the samples just didn't contain fat which to my understanding then changes the interpretation a lot. Further, in the figure legend of Appendix Figure S1 it says that three samples of cirrhotic livers are shown. However, sample c (and again k in Appendix figure S3) is not cirrhotic, does not even display advanced fibrosis but just portal fibrosis. Interestingly, in the main text of the manuscript it was described correctly (see line 93-95). Was this just a mistake or did the authors work with sample C as cirrhotic liver? Again, given the low number of samples this can change the interpretation a lot. Further, I don't think that the HIF1alpha staining in figure 7 worked very well. Normally HIF 1alpha isn't that patchy but more diffuse. Did the authors include proper controls or how can they assume that this staining really worked?

Further for large parts of the manuscript, a validation on the protein level is unfortunately missing. The authors present some data e.g. on SOX9 (immunohistochemistry) which is great but for several other exciting findings (such as atypical HC1 showing overlapping gene signatures with CC1 in Fig 5 D, E or Appendix Figure S10 A) it would be absolutely necessary to have a validation on the protein level e.g. by immunohistochemistry or immunofluorescence to understand the biological significance. Same for Figure 1 G,K - here, it would be very nice to highlight the findings in an additional cytokeratin 7 immunostaining to show the bile ducts and their morphology as bile duct hyperplasia is a common finding in cirrhotic liver and nothing new.

I think all these issues need to be clarified as this work might probably be considered as a resource study with a high likelihood to get cited. Therefore I don't think the manuscript given in its present form is appropriate for publication.

We appreciate your time and effort in reviewing our manuscript. We have taken the editors' comments into context while we respond to your primary concerns below.

1. Histological Expertise and Sample Representation

We acknowledge the importance of histological expertise in integrating spatial genetic data with morphological features. For this reason, our study is guided by the clinical service in Manchester and therefore expert clinical liver histopathologists as part of the Manchester Foundation Trust, who histologically confirmed fibrosis and cirrhosis in our samples (as detailed in Methods, Human Tissue Section & Table EV1).

Regarding steatosis in MASLD and ALD patients: While fatty liver is a hallmark of both alcohol and MASLD, progressive disease and cirrhosis often result in the abrogation of steatosis due to hepatocyte loss and fibrosis. For the experimental cohort, our study specifically examined patients with early cirrhosis (Child-Pugh A), where fibrosis predominates over steatosis (see Methods, Human Tissue Section and Table EV1). Critically, as part of a clinical service, clearly we would not have the same tissue the surgery team provide the pathologist as part of the clinical care pathway. We have clarified this in the methods. For the validation cohort, with a MASLD background, the H&E's from the full biopsy samples show aspects of steatosis for all samples (Appendix Figure S8).

2. Classification of Cirrhotic Samples

We appreciate the attention to sample classification. The main text (lines 93-95; now lines 99-102) correctly describes the observed differences in scarring. Importantly, while pathology obtains resected tissue as part of the clinical care pathway, we can only process fresh tissue (described in our manuscript) and as a result have very little control over the tissue we retrieve from surgery. However, all samples are from patients clinically diagnosed with liver cirrhosis (as described in table EV1) and this discrepancy in the figure legend has been corrected. We do not feel this impacts our overall findings, as the molecular data were analyzed appropriately for fibrosis progression (the reviewer acknowledges our technical excellence).

3. HIF1alpha Staining in Figure 7

We understand the concerns about HIF1alpha staining patterns. Variability in staining can arise due to differences in tissue processing, hypoxia gradients, and antigen retrieval techniques. All IMC antibodies are independently assessed as part of the core facility. In the literature there is evidence for HIF1alpha staining as both diffuse and patchy depending on tissue hypoxia dynamics (particularly in cancer / HCC), and our findings align with these known variations. Indeed, **focal overexpression** may be seen in specific regions with severe hypoxia or intense fibrogenic activity, particularly around fibrotic septa and perivenular areas.

4. Validation at the Protein Level

While we agree that protein-level validation strengthens findings, we emphasize that our conclusions are based on multi-modal integration of transcriptomics and chromatin accessibility—a powerful and impactful approach. While additional immunostaining could further support this, obtaining suitable antibodies (as the reviewer has already indicated above) are critical and whilst we have invested in this already as follow up work many have proved inadequate. We do not think this detracts from the current findings (also acknowledged by the other two reviewers) and additional experiments would be required to suitably reflect biological significance (similar to our data presented in Appendix Figures S22-S24).

Regarding Figure 1G, K and bile duct hyperplasia: We acknowledge that bile duct hyperplasia is a well-documented feature of cirrhosis. However, our study highlights the altered

molecular signatures of cell states at the scar interface, which is the novel aspect. CK7 staining would be complementary, but our focus was on transcriptional and regulatory changes, not just morphological confirmation. As the reviewer points out, the addition of CK7 would add little to these data.

In conclusion, we thank the reviewer for their detailed feedback. We have made any necessary textual revisions for clarification and robustness of our findings. Given the high technical quality and integrative nature of our study, we feel it serves as a valuable resource in liver fibrosis research and, as the reviewer points out, has a 'high likelihood to get cited'.

Referee #3 (Remarks for Author):

I am satisfied with the answers to my concerns. I believe that the results, especially those describing the transition states and the molecular signatures associated with them in the fibrotic niche, will have a significant impact for the field.

Thank you for your time assessing our manuscript. We are very grateful.

25th Mar 2025

Dear Prof. Piper Hanley,

We are pleased to inform you that your manuscript is accepted for publication and is now being sent to our publisher to be included in the next available issue of EMBO Molecular Medicine.
